# NEAR-OPTIMAL SOLUTIONS OF CONSTRAINED LEARNING PROBLEMS

**Juan Elenter** *
University of Pennsylvania

**Luiz F. O. Chamon**
University of Stuttgart

**Alejandro Ribeiro**
University of Pennsylvania

## ABSTRACT

With the widespread adoption of machine learning systems, the need to curtail their behavior has become increasingly apparent. This is evidenced by recent advancements towards developing models that satisfy robustness, safety, and fairness requirements. These requirements can be imposed (with generalization guarantees) by formulating constrained learning problems that can then be tackled by dual ascent algorithms. Yet, though these algorithms converge in objective value, even in non-convex settings, they cannot guarantee that their outcome is feasible. Doing so requires randomizing over all iterates, which is impractical in virtually any modern applications. Still, final iterates have been observed to perform well in practice. In this work, we address this gap between theory and practice by characterizing the constraint violation of Lagrangian minimizers associated with optimal dual variables, despite lack of convexity. To do this, we leverage the fact that non-convex, finite-dimensional constrained learning problems can be seen as parametrizations of convex, functional problems. Our results show that rich parametrizations effectively mitigate the issue of feasibility in dual methods, shedding light on prior empirical successes of dual learning. We illustrate our findings in fair learning tasks.

## 1 INTRODUCTION

Machine learning (ML) has become a core technology of information systems, reaching critical applications from medical diagnostics (Engelhard et al., 2023) to autonomous driving (Kiran et al., 2021). Consequently, it has become paramount to develop ML systems that not only excel at their main task, but also adhere to requirements such as fairness and robustness.

Since virtually all ML models are trained using empirical risk minimization (ERM) (Vapnik, 1999), a natural way to impose requirements is to explicitly add constraints to these optimization problems (Cotter et al., 2018; Chamon & Ribeiro, 2020; Velloso & Van Hentenryck, 2020; Fioretto et al., 2021; Chamon et al., 2023). Recent works (Chamon & Ribeiro, 2020; Chamon et al., 2023) have shown that from a probably approximately correct (PAC) perspective, constrained learning is essentially as hard as classical learning and that, despite non-convexity, it can be tackled using *dual algorithms* that only involve a sequence of regularized, unconstrained ERM problems. This approach has been used in several domains, such as federated learning (Shen et al., 2022), fairness (Cotter et al., 2019; Tran et al., 2021), active learning (Elenter et al., 2022), adversarial robustness (Robey et al., 2021), and data augmentation (Hounie et al., 2022).

These theoretical works, however, only address (i) the *estimation error*, arising from the empirical approximation of expectations in ERM and (ii) the *approximation error*, arising from using finite-dimensional models with limited functional representation capability. These are the leading challenges in unconstrained learning since the convergence properties of unconstrained optimization algorithms are well-understood in convex (e.g., (Bertsekas, 1997; Boyd & Vandenberghe, 2004)) as well as many non-convex settings (e.g., for overparametrized models (Soltanolkotabi et al., 2018; Brutzkus & Globerson, 2017; Ge et al., 2017)). This is not the case in constrained learning, where (iii) the *optimization error* can play a crucial role.

---

*Corresponding author: elenter@seas.upenn.edu

Indeed, dual methods are severely limited when it comes to recovering feasible solutions for constrained problems. In fact, not only might their primal iterates not converge to a feasible point [e.g, Fig. 1 or (Cotter et al., 2019, Section 6.3.1)], but they might not converge at all, displaying a cyclostationary behavior instead. This problem is hard even from an algorithmic complexity point-of-view (Daskalakis et al., 2021). For convex problems, this issue can be overcome by simply averaging the iterates (Nedić & Ozdaglar, 2009). Non-convex problems, however, require randomization (Kearns et al., 2018; Agarwal et al., 2018; Goh et al., 2016; Chamon et al., 2023). This approach is not only impractical, given the need to store a growing sequence of primal iterates, but also raises ethical considerations, since randomization further hinders explainability.

Yet, it has been observed that for typical modern ML tasks, taking the last or best iterate can perform well in practice (Cotter et al., 2018; Chamon & Ribeiro, 2020; Chamon et al., 2023; Robey et al., 2021; Elenter et al., 2022; Hounie et al., 2022; Shen et al., 2022; Gallego-Posada et al., 2022). This work addresses this gap between theory and practice by characterizing the sub-optimality and infeasibility of primal solutions associated with optimal dual variables. To do so, we observe that, though non-convex, constrained learning problems are generally parametrized versions of benign functional optimization problems. We then show that for sufficiently rich parametrizations, solutions obtained by dual algorithms closely approximate these functional solutions, not only in terms of optimal value as per (Cotter et al., 2019; Chamon & Ribeiro, 2020; Chamon et al., 2023), but also in terms of constraint satisfaction. This implies that dual ascent methods yield near-optimal and near-feasible solutions without randomization, despite non-convexity.

## 2 CONSTRAINED LEARNING

### 2.1 STATISTICAL CONSTRAINED RISK MINIMIZATION

As in classical learning, constrained learning tasks can be formulated as a statistical optimization problem, namely,

$$P_p^\star = \min_{\theta \in \Theta} \quad \ell_0(f_\theta) := \mathbb{E}_{(x,y)}[\tilde{\ell}_0(f_\theta(x), y)]$$
$$\text{s. to } \ell_i(f_\theta) := \mathbb{E}_{(x,y)}[\tilde{\ell}_i(f_\theta(x), y)] \le 0, \quad i = 1, .., m \tag{$P_p$}$$

where $f_\theta : \mathcal{X} \to \mathcal{Y}$ is a function associated with the parameter vector $\theta \in \Theta \subseteq \mathbb{R}^p$ and the hypothesis class $\mathcal{F}_\theta = \{f_\theta : \theta \in \Theta\}$ induced by this family of functions is assumed to be a subset of some compact functional space $\mathcal{F} \subset L^2$. Throughout the paper, we use the subscript $p$ (*parametrized*) to refer to quantities related to $(P_p)$. The functionals $\ell_i : \mathcal{F} \to \mathbb{R}$, $i = 0, .., m$, denote expected risks for loss functions $\tilde{\ell}_i$. In this setting, $\ell_0$ can be interpreted as a top-line metric (e.g., accuracy), while the functionals $\ell_1, .., \ell_m$ encode statistical requirements that the solution must satisfy (see example below).

**Example 2.1: Learning under counterfactual fairness constraints.** Consider the problem of learning an accurate classifier that is insensitive to changes in a set of protected attributes. Due to the correlation between these attributes and other features, simply hiding them from the model is not enough to guarantee this insensitivity. To do so, this requirement must be enforced explicitly. Indeed, consider the COMPAS study (ProPublica, 2020), with the goal of predicting recidivism based on past offense data while controlling for gender and racial bias. Explicitly, let $\tilde{\ell}_0$ denote the cross-entropy loss $\tilde{\ell}_0(\hat{y}, y) = -\log[\hat{y}]_y$. By collecting the protected features into the separate vector $z$, i.e., $x = [\tilde{x}, z]$, we can formulate the problem of learning a predictor insensitive to transformations $\rho_i$ that encompass all possible single variable modifications of $z$. Explicitly,

$$\min_{\theta \in \mathbb{R}^p} \mathbb{E}\left[\tilde{\ell}_0\left(f_\theta(x), y\right)\right]$$
$$\text{s. to } \mathbb{E}\left[D_{\text{KL}}(f_\theta(\tilde{x}, z) \,\|\, f_\theta(\tilde{x}, \rho_i(z)))\right] \le c, \quad i = 1, \ldots, m,$$

where $c > 0$ is the desired sensitivity level. Note that this formulation corresponds to the notion of (average) counterfactual fairness from (Kusner et al., 2018, Definition 5). In this setting, each constraint represents a requirement that the output of the classifier be *near-invariant* to changes in the protected features (here, gender and race). For instance, the prediction should be (almost) the

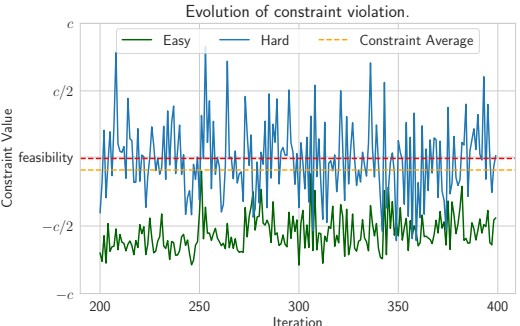 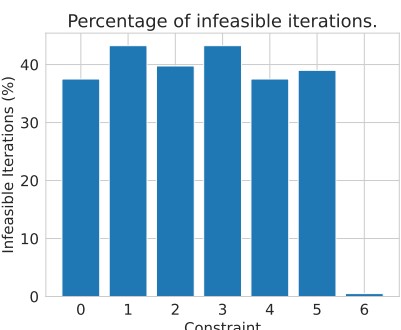

Figure 1: Feastibility of primal iterates in a constrained learning problem with fairness requirements. **Left**: Example of a hard constraint which oscillates between feasibiliy and infeasibility, and an easy constraint which remains feasible for all iterations. **Right**: After training accuracy has settled (around half of training epochs), all but the last constraint are infeasible 30-45 % of the iterations. In fact, at least one constraint is violated on 85% of the iterations shown. We cannot therefore stop the algorithm and expect to obtain a feasible solution.

same whether, all else being equal, the gender of the input is changed from "Male" to "Female" (and vice-versa) or the race is changed from "Caucasian" to "African-American."

Note that even if the losses $\tilde{\ell}_i(\hat{y}, y)$ are convex in $(\hat{y}, y)$ (as is the case of the cross-entropy), the functions $\ell_i$ need not be convex in $\theta$. This is the case, for instance, for typical modern ML models (e.g., if $f_\theta$ is a neural network, NN). Hence, $(P_p)$ is usually a non-convex optimization problem for which there is no straightforward way to project onto the feasibility set (e.g., onto the set of fair NNs). In light of these challenges, we turn to Lagrangian duality.

## 2.2 LEARNING IN THE DUAL DOMAIN

Let the Lagrangian $L : \mathcal{F} \times \mathbb{R}_+^m \to \mathbb{R}$ be defined as

$$L(\phi, \lambda) = \ell_0(\phi) + \lambda^T \ell(\phi), \tag{1}$$

where $\ell = [\ell_1, .., \ell_m]$ is a vector-valued functional collecting the constraints of $(P_p)$. For reasons that will become apparent later, we define $L$ over $\mathcal{F} \supseteq \mathcal{F}_\theta$. For a fixed dual variable $\lambda_p$, the Lagrangian $L(f_\theta, \lambda_p)$ is a regularized version of $(P_p)$, where $\ell$ acts as the regularizing functional. This leads to the dual function

$$g_p(\lambda_p) = \min_{\theta \in \Theta} L(f_\theta, \lambda_p), \tag{2}$$

based on which we can in turn define the dual problem of $(P_p)$ as

$$D_p^\star = \max_{\lambda_p \succeq 0} g_p(\lambda_p). \tag{$D_p$}$$

This saddle-point problem can be viewed as a two-player game or as a regularized learning problem, where the regularization parameter is also an optimization variable. As such, $(D_p)$ is a relaxation of $(P_p)$, implying that $D_p^\star \leq P_p^\star$. This is known as weak duality (Bertsekas, 1997).

The dual function $g_p$ in (2) is concave, irrespective of whether $(P_p)$ is convex (it is the pointwise minimum of a family of affine functions on $\lambda$ (Boyd & Vandenberghe, 2004)). As such, though $g_p$ may not be differentiable, it can be equipped with *supergradients* that provide potential ascent directions. Explicitly, a vector $s \in \mathbb{R}^m$ is a supergradient of the concave function $h : \mathbb{R}^m \to \mathbb{R}$ at a point $x$ if $h(z) - h(x) \geq s^T(z - x)$ for all $z$. The set of all supergradients of $h$ at $x$ is called the *superdifferential* and is denoted $\partial h(x)$. When the losses $\ell_i$ are continuous, the superdifferential of $g_p$ admits a simple description (Nedić & Ozdaglar, 2009), namely,

$$\partial g_p(\lambda_p) = \text{conv}\big[\ell(f_\theta(\lambda_p)) : f_\theta(\lambda_p) \in \mathcal{F}_\theta^\star(\lambda_p)\big],$$

where $\text{conv}(\mathcal{S})$ denotes the convex hull of the set $\mathcal{S}$ and $\mathcal{F}_\theta^\star(\lambda_p)$ denotes the set of Lagrangian minimizers $f_\theta(\lambda_p)$ associated to the multiplier $\lambda_p$, i.e.,

$$\mathcal{F}_\theta^\star(\lambda_p) = \operatorname*{arg\,min}_{\theta \in \Theta} L(f_\theta, \lambda_p). \tag{3}$$

---

**Algorithm 1** Dual Constrained Learning

---

1: *Inputs*: number of iterations $T \in \mathbb{N}$, step size $\eta > 0$.
2: *Initialize*: $\lambda(1) = 0$
3: **for** $t = 1, \ldots, T$ **do**
4:     Obtain $f_\theta(t)$ such that

$$f_\theta(t) \in \underset{\theta \in \Theta}{\arg\min}\ \ell_0(f_\theta) + \lambda(t)^T \ell(f_\theta) = \underset{\theta \in \Theta}{\arg\min}\ L(f_\theta, \lambda(t))$$

5:     Update dual variables

$$\lambda_i(t+1) = \max\left[0, \lambda_i(t) + \eta\, \ell_i(f_\theta(t))\right]$$

6: **end for**

---

In particular, this leads to an algorithm for solving $(D_p)$ known as projected supergradient ascent (Polyak, 1987; Shor, 2013) that we summarize in Algorithm 1.

When executing Algorithm 1, dual iterates $\lambda_p(t)$ move in ascent directions of the concave function $g_p$ (Shor, 2013, Section 2.4). Yet, the sequence of primal iterates $\{f_\theta(t)\}_{t=1}^T$ obtained as a by-product need not approach the set of solutions of $(P_p)$. The experiment in Figure 1 showcases this behaviour and illustrates that, in general, one can not simply stop the dual ascent algorithm at any iteration $t$ and expect the primal iterate $f_\theta(t)$ to be feasible. Additionally, the Lagrangian minimizers are not unique. In particular, for an optimal dual variable $\lambda_p^\star \in \Lambda_p^\star$, the set $\mathcal{F}_\theta^\star(\lambda_p^\star)$ is typically not a singleton and could contain infeasible elements (i.e, $\ell_i(f_\theta(\lambda_p^\star)) > 0$ for some $i \geq 1$). Even more so, as $\lambda_p(t)$ approaches $\Lambda_p^\star$, the constraint satisfaction of primal iterates can exhibit pathological cyclostationary behaviour, where one or more constraints oscillate between feasibility and infeasibility, see e.g., (Cotter et al., 2019, Section 6.3.1). For these reasons, convergence guarantees for non-convex optimization problems typically require randomization over (a subset of) the sequence $\{f_\theta(t)\}_{t=1}^T$, which is far from practical [see e.g, (Agarwal et al., 2018, Theorem 2), (Kearns et al., 2018, Theorem 4.1), (Cotter et al., 2019, Theorem 2), (Chamon et al., 2023, Theorem 3)]. In the sequel, we show conditions under which this is not necessary.

## 3   NEAR-OPTIMAL SOLUTIONS OF CONSTRAINED LEARNING PROBLEMS

Primal iterates obtained as a by-product of the dual ascent method in Algorithm 1 may fail to be solutions of $(P_p)$. However, it has been observed that taking the last or best iterate can perform well in practice. This can be understood by viewing $(P_p)$ as the parametrized version of a benign functional program, ammenable to a Lagrangian formulation. This *unparametrized* problem does not suffer from the same limitations as $(P_p)$ in terms of primal recovery and we can thus use its solution as a reference point to measure the sub-optimality of the primal iterates obtained with Algorithm 1.

The *unparametrized* constrained learning problem is defined as

$$\begin{aligned} P_u^\star = \min_{\phi \in \mathcal{F}} \quad & \ell_0(\phi) \\ \text{s.to}\ \ & \ell_i(\phi) \leq 0, \quad i = 1, .., m \end{aligned} \tag{$P_u$}$$

where $\mathcal{F}$ is a convex, compact subset of an $L^2$ space. For instance, $\mathcal{F}$ can be a subset of the space of continuous functions or a reproducing kernel Hilbert space (RKHS) and $\mathcal{F}_\theta$ can be induced by a neural network architecture with smooth activations or a finite linear combinations of kernels. In both cases, we know that $\mathcal{F}_\theta$ can uniformly approximate $\mathcal{F}$ arbitrarily well as the dimension of $\theta$ grows (Hornik, 1991; Berlinet & Thomas-Agnan, 2011). The smallest choice of $\mathcal{F}$ is in fact $\overline{\text{conv}}(\mathcal{F}_\theta)$ (closed convex hull of $\mathcal{F}_\theta$).

Analogous to the definitions from Section 2.1,

$$g_u(\lambda_u) := \min_{\phi \in \mathcal{F}} L(\phi, \lambda_u)$$

denotes the unparametrized dual function, $\mathcal{F}^{\star}(\lambda_u) = \arg\min_{\phi \in \mathcal{F}} L(\phi, \lambda_u)$ is the set of Lagrangian minimizers $\phi(\lambda_u)$ associated to $\lambda_u$ and

$$D_u^{\star} = \max_{\lambda_u \succeq 0} g_u(\lambda_u) \qquad (D_u)$$

is the unparametrized dual problem. The subscript $u$ is used to denote quantities related to the unparametrized problem $(P_u)$. We now present two assumptions that allow us to characterize the relation between the dual and primal solutions of problem $D_u$.

**Assumption 3.1.** *The functionals $\ell_i$ , $i = 0, \dots, m$, are convex and $M-$Lipschitz continuous in $\mathcal{F}$. Additionally, $\ell_0$ is $\mu_0-$strongly convex.*

Note that we require convexity of the losses with respect to their functional arguments and not model parameters $\theta$, which holds for most typical losses, e.g, mean squared error and cross-entropy loss.

**Assumption 3.2.** *There exists $\phi \in \mathcal{F}$ such that $\ell(\phi) \prec \min[\mathbf{0}, \ell(\phi(\lambda_p^{\star})), \ell(f_\theta(\lambda_p^{\star}))]$ for all $\phi(\lambda_p^{\star}) \in \mathcal{F}(\lambda_p^{\star})$, $f_\theta(\lambda_p^{\star}) \in \mathcal{F}_\theta(\lambda_p^{\star})$ and $\lambda_p^{\star} \in \Lambda_p^{\star}$.*

Assumption 3.2 is a stronger version of Slater's constraint qualification, which requires only $\ell(\phi) \prec \mathbf{0}$. Here, we require the existence of a (suboptimal) candidate $\phi$ that is strictly feasible even for perturbed versions of $(P_u)$.

Under these assumptions, the Lagrangian minimizer is unique. This makes the superdifferential of the dual function a singleton at every $\lambda_u$: $\partial g_u(\lambda_u) = \{\ell(\phi(\lambda_u))\}$, which means that the dual function $g_u(\lambda_u)$ is differentiable (Shor, 2013). Let $\phi^{\star}$ be a solution of problem $(P_u)$. Assumptions 3.1 and 3.2 imply that strong duality (i.e, $P_u^{\star} = D_u^{\star}$) holds in this problem, and that at $\lambda_u^{\star}$, there is a unique Lagrangian minimizer $\phi^{\star}(\lambda_u^{\star}) = \phi^{\star}$ which is, by definition, feasible (Bertsekas, 1997).

The only difference between problems $(P_p)$ and $(P_u)$ is the set over which the optimization is carried out. Thus, if the parametrization $\Theta$ is rich enough (e.g, deep neural networks), the set $\mathcal{F}_\theta$ is essentially the same as $\mathcal{F}$, and we should expect the properties of the solutions to problems $(D_p)$ and $(D_u)$ to be similar. This insight leads us to the $\nu-$near universality of the parametrization assumption.

**Assumption 3.3.** *For all $\phi \in \mathcal{F}$, there exists $\theta \in \Theta$ such that $\|\phi - f_\theta\|_{L_2} \leq \nu$.*

The constant $\nu$ in Assumption 3.3 is a measure of how well $\mathcal{F}_\theta$ covers $\mathcal{F}$. Consider, for instance, that $\mathcal{F}$ is the set of continuous functions and $\mathcal{F}_\theta$ the set of functions implementable with a two-layer neural network with sigmoid activations and $K$ hidden neurons. If the parametrization has 10 neurons in the hidden layer, it is considerably worse at representing elements in $\mathcal{F}$ than one with 1000 neurons. While determining the exact value of $\nu$ is in general not straightforward, any $\nu > 0$ can be achieved for a large enough number of neurons (Hornik, 1991). The same holds for the number of kernels and an RKHS (Berlinet & Thomas-Agnan, 2011).

Given these facts, it is legitimate to ask: how close are the elements of $\mathcal{F}_\theta(\lambda_p^{\star})$ to $\phi^{\star}$ in terms of their optimality *and* constraint satisfaction? Bounding these errors would theoretically justify the use of last primal iterates, doing away with the need for randomization.

### 3.1 NEAR-OPTIMALITY AND NEAR-FEASIBILITY OF DUAL LEARNING

A key challenge of using duality to undertake $(P_p)$ is that the value $D_p^{\star}$ of the dual problem $(D_p)$ need not be a good approximation of the value $P_p^{\star}$ of $(P_p)$ (i.e, lack of strong duality). This was tackled in (Chamon et al., 2023, Prop. 3.3). Explicitly, under Assumptions 3.1-3.3, the duality gap of problem $(P_p)$ is bounded as in

$$P_p^{\star} - D_p^{\star} \leq M\nu(1 + \|\tilde{\lambda}^{\star}\|_1) := \Gamma_1, \qquad (4)$$

where $\tilde{\lambda}^{\star}$ maximizes $\tilde{g}_p(\lambda) = g_p(\lambda) + M\nu\|\lambda\|_1$. This result, however, only shows that the dual problem can be used to approximate the *value* of the constrained problem $(P_p)$. It says nothing about whether it can provide a (near-)feasible solution, which is the main issue addressed in this paper. We next characterize the sub-optimality and constraint violation of the Lagrangian minimizers $f_\theta(\lambda_p^{\star}) \in \mathcal{F}_\theta(\lambda_p^{\star})$ by comparing these primal variables with the solution of the unparametrized problem $\phi^{\star}$.

The curvature of the unparametrized dual function $g_u(\lambda_u)$ around its optimum is central to this analysis. We will first provide a result with the following assumption on this curvature and then

describe its connection to the properties of $(P_p)$. Let $\mathcal{H}_\lambda := \{\gamma\lambda_u^\star + (1-\gamma)\lambda_p^\star : \gamma \in [0,1]\}$ denote the segment connecting $\lambda_u^\star$ and $\lambda_p^\star$.

**Assumption 3.4.** *The dual function $g_u$ is $\mu_g-$strongly concave and $\beta_g-$smooth along $\mathcal{H}_\lambda$.*

The following proposition characterizes the constraint violation for all $f_\theta(\lambda_p^\star) \in \mathcal{F}_\theta^\star(\lambda_p^\star)$ with respect to $\phi^\star$; the optimal, feasible solution of the unparametrized problem.

**Proposition 3.1.** *Under Assumptions 3.1-3.4, any $f_\theta(\lambda_p^\star) \in \mathcal{F}_\theta^\star(\lambda_p^\star)$, approximates the constraint value of the solution $\phi^*$ of $(P_u)$ as in:*

$$\|\ell(f_\theta(\lambda_p^\star)) - \ell(\phi^\star)\|_2^2 \leq 2\beta_g M\nu(1 + \|\lambda_p^\star\|_1)\left(1 + \sqrt{\frac{\beta_g}{\mu_g}}\right)^2.$$

Since $(P_u)$ is feasible, $\ell(\phi^\star)$ is non-positive. Hence, the approximation bound in Proposition 3.1 is stronger than an infeasibility bound on $f_\theta(\lambda_p^\star)$. Indeed, it says not that $\ell(f_\theta(\lambda_p^\star)) \leq 0$, but that it approximates the constraint values of the optimal solution $\phi^\star$. The ratio $\beta_g/\mu_g$ (i.e, the condition number of $g_u$), which determines optimal step sizes in dual ascent methods (Polyak, 1987), also plays a key role here, representing the tension between two fundamental forces driving this bound. On the one hand, the sensitivity of the dual problems, controlled by $\mu_g$, which determines how different $\lambda_u^\star$ and $\lambda_p^\star$ are. On the other hand, the sensitivity of the primal problems, linked to the smoothness constant $\beta_g$, which determines the effect of this difference on feasibility.

Nevertheless, Proposition 3.1 remains abstract. To connect it to the properties of $(P_p)$, we rely on the following assumptions to obtain bounds on $\mu_g$ and $\beta_g$.

**Assumption 3.5.** *The functionals $\ell_i$, $i = 0, \dots, m$ are $\beta$-smooth on $\mathcal{F}$.*

**Assumption 3.6.** *The Jacobian $D_\phi\ell(\phi^\star)$ is full-row rank at the optimum, i.e, there exists $\sigma > 0$ such that $\inf_{\|\lambda\|_2=1} \|\lambda^T D_\phi\ell(\phi^\star)\|_{L_2} \geq \sigma$, where $D_\phi\ell(\phi^\star)$ denotes the Fréchet derivative of the functional $\ell$ at $\phi^\star$ (see definition in Appendix A.1).*

Assumption 3.6 is unlike the previous regularity assumptions over which a practitioner has full control and is not straightforward to satisfy at first sight. It is, however, a typical assumption used to derive duality results in convex optimization known as linear independence constraint qualification or LICQ (Bertsekas, 1997). As such, it could be replaced by a different constraint qualification, such as a stricter version of Assumption 3.2. This is, however, left for future work. Under these assumptions, we can describe the curvature of $g_u$ in terms of the problem parameters as follows.

**Lemma 3.1.** *Under assumptions 3.1, 3.2, 3.5 and 3.6, $g_u(\lambda_u)$ is $\mu_g-$strongly concave and $\beta_g-$smooth on $\mathcal{H}_\lambda$ for $\mu_g = \frac{\mu_0\,\sigma^2}{\beta^2(1+\Delta)^2}$ and $\beta_g = \frac{\sqrt{m}M^2}{\mu_0}$, where $\Delta = \max(\|\lambda_u^\star\|_1, \|\lambda_p^\star\|_1)$.*

Having characterized the curvature of the unparametrized dual function $g_u$, we can now state the main result of this section, which puts together Proposition 3.1, Lemma 3.1, and the near-optimality result from (Chamon et al., 2023) in (4) to bound the near-optimality *and* near-feasibility of Lagrangian minimizers associated to optimal dual variables.

> **Theorem 3.1.** *Under assumptions 3.1, 3.2, 3.3, 3.5 and 3.6, the sub-optimality and infeasibility of any $f_\theta(\lambda_p^\star) \in \mathcal{F}(\lambda_p^\star)$ is bounded by:*
>
> $$\|\ell(f_\theta(\lambda_p^\star)) - \ell(\phi^\star)\|_\infty \leq \Gamma_2 := M\left[1 + \kappa_1\kappa_0(1+\Delta)\right]\sqrt{2m\frac{M\nu}{\mu_0}(1 + \|\lambda_p^\star\|_1)} \quad (5)$$
>
> $$|P_p^\star - \ell_0(f_\theta(\lambda_p^\star))| \leq (1 + \|\lambda_p^\star\|_1)M\nu + \Gamma_1 + \|\lambda_p^\star\|_1\Gamma_2 \quad (6)$$
>
> *with $\kappa_1 = \frac{M}{\sigma}$, $\kappa_0 = \frac{\beta}{\mu_0}$, $\Delta = \max\{\|\lambda_u^\star\|_1, \|\lambda_p^\star\|_1\}$ and $\Gamma_1$ as in (4).*

Theorem 3.1 shows that the dual problem $(D_p)$ not only approximates the value $P_p^\star$ of $(P_p)$, but also provides approximate solutions for it. The quality of these approximations depends on three factors. First, the sensitivity of the learning problem, as captured by the Lipschitz constant $M$ and the constants $\kappa_1$ and $\kappa_0$, that correspond to the condition numbers of the constraint Jacobian and the

objective function respectively. Overall, these quantities measure how well-conditioned the problem is. Second, the requirements difficulty. Indeed, the optimal dual variables can be seen as measures of the sensitivity of the objective value with respect to constraint perturbations (see, e.g., (Boyd & Vandenberghe, 2004)). Hence, the more stringent the constraints, the larger $\|\lambda_u^\star\|_1$ and/or $\|\lambda_p^\star\|_1$.

Finally, the approximation error depends on the factor $\nu$ that denotes the richness of the parametrization, i.e., how good it is at approximating functions in $\mathcal{F}$ (Assumption 3.3). In fact, Theorem 3.1 shows that as the model capacity increases ($\nu$ decreases), the approximation bounds (5)–(6) improve. This behavior is not trivial. Indeed, while we expect that richer parametrizations lead to lower approximation errors, Theorem 3.1 states that they also make solving the optimization problem $(P_p)$ easier, since dual solutions then provide better approximations of primal solutions. Observe that the effect of these factors on feasibility in (5) are similar to those on optimality in (6) and, e.g., (Chamon et al., 2023). Next, we leverage these results to provide convergence guarantees for Algorithm 1. But first, we outline the main ideas behind the proof of Proposition 3.1.

## 3.2 PROOF SKETCH

In this section, we provide a brief outline of the proof of Theorem 3.1. We begin by decomposing the distance between constraint violations as

$$
\begin{aligned}
\|\ell(f_\theta(\lambda_p^\star)) - \ell(\phi^\star)\|_2 &= \|\ell(f_\theta(\lambda_p^\star)) - \ell(\phi(\lambda_p^\star)) + \ell(\phi(\lambda_p^\star)) - \ell(\phi^\star)\|_2 \\
&\leq \|\ell(f_\theta(\lambda_p^\star)) - \ell(\phi(\lambda_p^\star))\|_2 + \|\ell(\phi(\lambda_p^\star)) - \ell(\phi(\lambda_u^\star))\|_2
\end{aligned}
\tag{7}
$$

The first term captures the effect of parametrizing the hypothesis class for a fixed dual variable. In contrast, the second term characterizes the effect of changing the dual variables on the unparametrized Lagrangian minimizer. This is made clear in (7) by using the fact that $\phi^\star = \phi(\lambda_u^\star)$ (see discussion in Section 3). In the sequel, we analyze each of these terms separately. For conciseness, all technical definitions from this section are deferred to Appendix A.1.

### 3.2.1 DUAL VARIABLE PERTURBATION

We begin by analyzing the second term in (7). Recall from the beginning of Section 3 that under Assumption 3.1–3.2, it holds that $\nabla_\lambda g_u(\lambda) = \ell(\phi(\lambda))$. Hence, $\|\ell(\phi(\lambda_p^\star)) - \ell(\phi(\lambda_u^\star))\|_2 = \|\nabla_\lambda g_u(\lambda_p^\star) - \nabla_\lambda g_u(\lambda_u^\star)\|_2$. Using the $\beta_g$-smoothness of $g_u$, this gradient difference can be bounded using $\|\lambda_p^\star - \lambda_u^\star\|_2$. The latter can in turn be bounded by combining the $\nu$-universality of the parametrization (Assumption 3.3) and convex optimization perturbation results to obtain:

**Proposition 3.2.** *Under assumptions 3.1-3.4, the distance between the constraint violations of $\phi(\lambda_p^\star)$ and $\phi(\lambda_u^\star)$ is bounded by:*

$$
\|\ell(\phi(\lambda_p^\star)) - \ell(\phi(\lambda_u^\star))\|_2^2 \leq 2\frac{\beta_g^2}{\mu_g} M \nu (1 + \|\lambda_p^\star\|_1)
\tag{8}
$$

### 3.2.2 HYPOTHESIS CLASS PERTURBATION

Bounding the first term in (7) is less straightforward. To do so, we rely on the perturbation function of the unparametrized problem $(P_u)$, defined as

$$
\begin{aligned}
P_u^\star(\epsilon) = \min_{\phi \in \mathcal{F}} \quad &\ell_0(\phi) \\
\text{s.to} \quad &\ell(\phi) + \epsilon \preceq 0,
\end{aligned}
\tag{$P_\epsilon$}
$$

for some perturbation $\epsilon \in \mathbb{R}^m$. Intuitively, $P_u^\star(\epsilon)$ quantifies the impact on the objective value of modifying the constraint specifications by $\epsilon$. Note that the unparametrized problem $(P_u)$ is recovered for $\epsilon = 0$. Motivated by the fact that we can get a strong handle on the sensitivity of the perturbation function $(P_\epsilon)$, we seek to bound $\|\ell(f_\theta(\lambda_p^\star)) - \ell(\phi(\lambda_p^\star))\|_2$ by instead analyzing $|P_u^\star(\epsilon_p) - P_u^\star(\epsilon_u)|$ for $\epsilon_p = -\ell(f_\theta(\lambda_p^\star))$ and $\epsilon_u = -\ell(\phi(\lambda_p^\star))$. Indeed, it holds for every $\lambda \in \mathbb{R}_+^m$ that $P^\dagger(\lambda) = -g_u(\lambda)$, where $^\dagger$ denotes the Fenchel conjugate (see Appendix A.4). We can therefore relate the curvature of $g_u$ to that $P_u^\star(\epsilon)$ (Kakade et al., 2009) to obtain:

**Proposition 3.3.** *Under assumptions 3.1-3.4, the distance between constraint violations associated to the parametrization of the hypothesis class is given by:*

$$
\|\ell(\phi(\lambda_p^\star)) - \ell(f_\theta(\lambda_p^\star))\|_2^2 \leq 2\beta_g M \nu (1 + \|\lambda_p^\star\|_1)
$$

Using Propositions 3.2–3.3 in (7) yields Proposition 3.1. Theorem 3.1 is then obtained by further leveraging Lemma 3.1 and the bound on the duality gap $P_p^\star - D_p^\star$ in (4) (see Appendix A.13.2).

## 4 BEST ITERATE CONVERGENCE

In this section, we leverage the connection between the parameterized [cf. $(P_p)$ and $(D_p)$] and unparameterized [cf. $(P_u)$ and $(D_u)$] problems to analyze the convergence of Algorithm 1. Seeking a more general result, we relax Steps 4 and 5 to allow for approximate Lagrangian minimization and the use of stochastic supergradients of the dual function respectively.

Explicitly, we assume that for all $t$, the oracle in Step 4 returns a function $f_\theta^o(t)$ such that

$$L(f_\theta^o(t), \lambda_p) \le \min_{\theta \in \Theta} L(f_\theta, \lambda(t)) + \rho, \tag{9}$$

for an approximation error $\rho \ge 0$. In contrast to Step 4, equation 9 accounts for potential numerical and approximation errors in the computation of the Lagrangian minimizer. The existence of such a $\rho$-approximate oracle is a typical assumption in the analysis of dual algorithms (Cotter et al., 2019; Chamon et al., 2023; Kearns et al., 2018) and is often justified by substantial theoretical and empirical evidence that many ML optimization problems can be efficiently solved despite non-convexity. That is the case, e.g., for deep neural networks (Zhang et al., 2021; Brutzkus & Globerson, 2017; Soltanolkotabi et al., 2018; Ge et al., 2017). Additionally, we consider that the dual variable update in Step 5 is replaced by

$$\lambda_i(t+1) = \max\left[0, \lambda_i(t) + \eta\, \hat{\ell}_i(f_\theta^o(t))\right], \tag{10}$$

where $\hat{\ell}_i(f_\theta^o(t))$ are conditionally unbiased estimates of the statistical risks $\ell_i(f_\theta^o(t))$, i.e., $\mathbb{E}[\hat{\ell}_i(f_\theta^o(t))|\lambda(t)] = \ell_i(f_\theta^o(t))$. This stochastic update accounts for, e.g., the use of *independent* sample batches to estimate the constraint slacks $\ell_i(f_\theta^o)$.

The following Lemma establishes the convergence of the best iterate of (9)–(10), i.e., of the dual variables $\lambda_i(t)$ that yield the largest dual function for all $t$.

**Lemma 4.1.** *Let* $g_p^{\text{best}}(t|\lambda(t_0)) = \max_{s \in [t_0, t]} g_p(\lambda(s))$ *be the maximum value of the parametrized dual function up to time $t$. Then, for all $t_0 > 1$, it holds that*

$$\lim_{t \to \infty} g_p^{\text{best}}(t|\lambda(t_0)) \ge D_p^\star - \left(\frac{\eta S^2}{2} + \rho\right) \quad a.s.,$$

*where* $S^2 > \sum_{i=1}^m \mathbb{E}\left[|\hat{\ell}_i(f_\theta^o(t))|^2|\lambda(t)\right]$.

The existence of a finite $S^2$ is implied by the assumption that $\mathcal{F}_\theta \subseteq \mathcal{F} \subset L^2$. Lemma 4.1 implies that for any $\delta > 0$, there exists a finite $t^\star$ such that $\lambda(t^\star)$ achieves the value $D_p^\star - \left(\frac{\eta S^2}{2} + \rho + \delta\right)$. We denote this iterate $\lambda^{\text{best}}$. Note that the step size $\eta$ can be reduced so as to make $g_p^{\text{best}}$ arbitrarily close to $D_p^\star - \rho$ (asymptotically). In view of the bound on the duality gap $P_p^\star - D_p^\star$ in (4), Lemma 4.1 implies that $\lambda^{\text{best}}$ is near-optimal. Combine with the near-feasibility results from Section 3, we can also bound the constraint violations of the Lagrangian minimizer associated with $\lambda^{\text{best}}$.

---

**Proposition 4.1.** *Let $\lambda^{\text{best}}$ be any dual iterate that achieves $g_p(\lambda^{\text{best}}) \ge D_p^\star - (\eta S^2/2 + \rho)$. Suppose there exists $\phi \in \mathcal{F}$ such that $\ell(\phi) \prec \min\{0, \ell(\phi(\lambda^{\text{best}})), \ell(f_\theta(\lambda^{\text{best}}))\}$ and that Assumptions 3.1, 3.3, 3.5, and 3.6 hold. Then,*

$$\|\ell(\phi^\star) - \ell(f_\theta(\lambda^{\text{best}})))\|_2^2 \le 2\beta_g\left(M\nu(1 + \|\lambda^{\text{best}}\|_1) + \frac{\eta S^2}{2} + \rho\right)\left(1 + \sqrt{\frac{\beta_g}{\tilde{\mu}_g}}\right)^2$$

*where* $\tilde{\mu}_g = \frac{\mu_0\,\sigma^2}{\beta^2(1 + \max\{\|\lambda_u^\star\|_1, \|\lambda^{\text{best}}\|_1\})^2}$.

---

Reasonably, the bound in Proposition 4.1 is governed by the same terms as Theorem 3.1. Here, however, the bound is the loosened by the sub-optimality of $\lambda^{\text{best}}$ with respect to $\lambda_p^\star$.

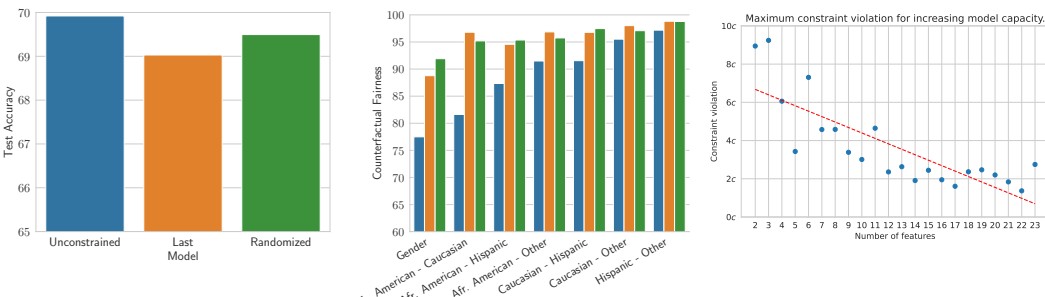

Figure 2: **Left:** the Unconstrained model performs better in terms of average test accuracy than both the Last and Randomized model. **Middle:** Both constrained models do better in terms of Counterfactual Fairness. The key point is that the Last iterate is never far from the Randomized one in terms of constraint violation. **Right:** As the richness of the parametrization increases the maximum constraint violation (i.e: size of the oscillations) decreases.

## 5    EXPERIMENTAL VALIDATION

To illustrate the theoretical results from Sections 3 and 4, we return to the counterfactually fair learning problem from Example 2.1. We work with the COMPAS dataset, where the task is to predict recidivism while remaining insensitive to the protected variables *gender* and *race*, which can take the values ["Male", "Female"] and ["African American", "Hispanic", "Caucasian", "Other"] respectively. We take the parametrized model $f_\theta$ to be a 2-layer NN with sigmoid activations, so that the resulting constrained learning problem is non-convex. Further experimental details are provided in Appendix A.16. We compare the accuracy and constraint satisfaction of three models: an *unconstrained* predictor, trained without any additional constraints; a *last* iterate predictor, corresponding to the final iterate $f_\theta(T)$ of an empirical version of Algorithm 1; and a *randomized* predictor that samples a model uniformly at random from the sequence of primal iterates $\{f_\theta(t)\}_{t=t_0}^T$ for each prediction.

As shown in Fig. 2 (**Left**), the unconstrained model is slightly better than the two constrained ones in terms of predictive accuracy. This advantage comes at the cost of less counterfactually fair predictions, i.e., a model more sensitive to the protected features (Fig. 2, **Middle**). The key point of this experiment, however, is that the *last iterate* and *randomized* predictors provide similar accuracy and constraint satisfaction, as predicted by Theorem 3.1. Additionally, Fig. 2 (**Right**) showcases the impact of the parametrization richness on the constraint violation of last primal iterates. We control this richness by means of projecting the data onto a lower dimensional space using a fixed, random linear map. Note that, as Theorem 3.1 indicates, the constraint violation decreases by up to an order of magnitude as we increase the capacity of the model. As we have observed before, this behavior is not straightforward: though richer parametrizations are expected to lead to lower approximation errors, it is not immediate that it should make the optimization problem $(P_p)$ easier to solve.

## 6    CONCLUSION

We analyzed primal iterates obtained from a dual ascent method when solving the Lagrangian dual of a primal non-convex constrained learning problem. The primal problem in question is the parametrized version of a convex functional program, which is amenable to a Lagrangian formulation. Specifically, we characterized how far these predictors are from the solution of the unparametrized problem in terms of their optimality and constraint violation. This result led to a characterization of the infeasibility of best primal iterates and elucidated the role of the capacity of the model and the curvature of the objective. These guarantees bridge a gap between theory and practice in constrained learning, shedding light on when and why randomization is unnecessary. The findings presented in this work can be extended in several ways. For instance, a study of the estimation error incurred by minimizing the empirical Lagrangian in Algorithm 1 could be added. It might also be possible to characterize the curvature of the dual function by alternative means, which could potentially lift assumptions on the unparametrized problem.

## 6.1 ACKNOWLEDGMENTS

The work of A. Ribeiro and J. Elenter is supported by NSF-Simons MoDL, Award 2031985, NSF AI Institutes program. The work of L.F.O. Chamon is funded by the Deutsche Forschungsgemeinschaft (DFG, German Research Foundation) under Germany's Excellence Strategy (EXC 2075-390740016).

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

# A    APPENDIX

## A.1    ADDITIONAL DEFINITIONS

**Definition A.1.** *We say that a functional $\ell_i : \mathcal{F} \to \mathbb{R}$ is* Fréchet differentiable *at $\phi^0 \in \mathcal{F}$ if there exists an operator $D_\phi \ell_i(\phi^0) \in \mathfrak{B}(\mathcal{F}, \mathbb{R})$ such that:*

$$\lim_{h \to 0} \frac{|\ell_i(\phi^0 + h) - \ell_i(\phi^0) - \langle D_\phi \ell_i(\phi^0), h \rangle|}{\|h\|_{L_2}} = 0$$

*where $\mathfrak{B}(\mathcal{F}, \mathbb{R})$ denotes the space of bounded linear operators from $\mathcal{F}$ to $\mathbb{R}$.*

The space $\mathfrak{B}(\mathcal{F}, \mathbb{R})$, algebraic dual of $\mathcal{F}$, is equipped with the corresponding dual norm:

$$\|B\|_{L_2} = \sup \left\{ \frac{|\langle B, \phi \rangle|}{\|\phi\|_{L_2}} \; : \; \phi \in \mathcal{F}, \, \|\phi\|_{L_2} \neq 0 \right\}$$

which coincides with the $L_2-$norm through Riesz's Representation Theorem: there exists a unique $g \in \mathcal{F}$ such that $B(\phi) = \langle \phi, g \rangle$ for all $\phi$ and $\|B\|_{L_2} = \|g\|_{L_2}$.

**Definition A.2.** *A function $h : \mathcal{X} \to \mathbb{R}$ is said to be closed if for each $\alpha \in \mathbb{R}$, the sublevel set $\{h(x) \leq \alpha \; : \; x \in \mathcal{X}\}$ is a closed set.*

**Definition A.3.** *A convex function $h : \mathcal{X} \to \mathbb{R}$ is proper if $h(x) > -\infty$ for all $x \in \mathcal{X}$ and there exists $x_0 \in \mathcal{X}$ such that $h(x_0) < +\infty$.*

**Definition A.4.** *Let $\mathcal{X}$ be an Euclidean vector space. Given a convex function $h : \mathcal{X} \to \mathbb{R} \cup \{\infty\}$, its Fenchel conjugate $h^\dagger : \mathcal{X} \to \mathbb{R} \cup \{\infty\}$ is defined as:*

$$h^\dagger(y) = \sup_{x \in \mathcal{X}} \langle x, y \rangle - h(x)$$

## A.2    PROOF OF LEMMA A.1: DISTANCE BETWEEN DUAL FUNCTIONS

**Lemma A.1.** *The point-wise distance between the parametrized and unparametrized dual functions is bounded by:*

$$0 \leq g_p(\lambda) - g_u(\lambda) \leq M\nu(1 + \|\lambda\|_1) \qquad \forall \quad \lambda \succeq 0 \tag{11}$$

As defined in section 2.1, $\phi(\lambda)$ denotes the Lagrangian minimizer associated to the multiplier $\lambda$ in the unparametrized problem.

By the near-universality assumption, $\exists \, \tilde{\theta} \in \Theta$ such that $\|\phi(\lambda) - f_{\tilde{\theta}}\|_{L_2} \leq \nu$. Note that,

$$L(f_{\tilde{\theta}}, \lambda) - L(\phi(\lambda), \lambda) = \ell_0(f_{\tilde{\theta}}) - \ell(\phi(\lambda)) + \lambda^T \left( \ell(f_{\tilde{\theta}}) - \ell(\phi(\lambda)) \right)$$

$$\leq \|\ell_0(f_{\tilde{\theta}}) - \ell(\phi(\lambda))\|_2 + \sum_{i=1}^{m} [\lambda]_i \|\ell(f_{\tilde{\theta}}) - \ell(\phi(\lambda))\|_2$$

where we used the triangle inequality twice. Then, using the $M-$Lipschitz continuity of the functionals $\ell_i$ and the fact that $\|\phi(\lambda) - f_{\tilde{\theta}}\|_2 \leq \nu$, we obtain:

$$L(f_{\tilde{\theta}}, \lambda) - L(\phi(\lambda), \lambda) \leq M \|f_{\tilde{\theta}} - \phi(\lambda)\|_{L_2} + M \sum_{i=1}^{m} [\lambda]_i \|f_{\tilde{\theta}} - \phi(\lambda)\|_{L_2}$$

$$\leq M\nu + M\nu \sum_{i=1}^{m} [\lambda]_i = M\nu(1 + \|\lambda\|_1)$$

Since $f_\theta(\lambda) \in \mathcal{F}_{\tilde{\theta}}^\star(\lambda)$ is a Lagrangian minimizer, we know that $L(f_\theta(\lambda), \lambda) \leq L(f_{\tilde{\theta}}, \lambda)$. Thus,

$$0 \leq L(f_\theta(\lambda), \lambda) - L(\phi(\lambda), \lambda) \leq L(f_{\tilde{\theta}}, \lambda) - L(\phi(\lambda), \lambda)$$

where the non-negativity comes from the fact that $\mathcal{F}_\Theta \subseteq \mathcal{F}$. This implies:

$$0 \leq g_p(\lambda) - g_u(\lambda) \leq M\nu(\|\lambda\|_1 + 1) \qquad \forall \quad \lambda \succeq 0$$

which conludes the proof.

## A.3 PROOF OF LEMMA A.2: DIFFERENTIABILITY OF $g_u(\lambda_u)$

**Lemma A.2.** *Under assumption 3.1, the unparametrized dual function $g_u(\lambda)$ is everywhere differentiable with gradient $\nabla_\lambda g_u(\lambda) = \ell(\phi(\lambda))$.*

From assumption 3.1, $\ell(\phi)$ is strongly convex and $\lambda^T \ell(\phi)$ is a non-negative combination of convex functions. Thus, the Lagrangian $L(f, \lambda)$ is strongly convex on $\phi$ for any fixed dual variable $\lambda \in \mathbb{R}^m_+$.

The convexity and compactness of $\mathcal{F}$ imply that, in the unparametrized problem, the Lagrangian functional attains its minimizer $\phi(\lambda)$ for each $\lambda$. (see e.g, (Kurdila & Zabarankin, 2006) Theorem 7.3.1.) Then, by the strong convexity of $L(\phi, \lambda)$, this minimizer is unique.

Since $L(f, \lambda)$ is affine on $\lambda$, it is differentiable on $\lambda$. Then, by application of the Generalized Danskin's Theorem (see e.g: (Başar & Bernhard, 2008) Corollary 10.1) to $g_u(\lambda)$ and using that the set of minimizers $\phi(\lambda)$ of $L(f, \lambda)$ is a singleton, we obtain:

$$\nabla_\lambda g_u(\lambda) = \ell(\phi(\lambda)),$$

which completes the proof.

## A.4 PROOF OF LEMMA A.3: DISTANCE BETWEEN OPTIMAL DUAL VARIABLES

**Lemma A.3.** *Under assumptions 3.1, 3.2, 3.3, 3.4, the proximity between the unparametrized and parametrized optimal dual variables is characterized by:*

$$\|\lambda_p^\star - \lambda_u^\star\|_2^2 \leq 2\frac{M\nu}{\mu_g}(1 + \|\lambda_p^\star\|_1) \tag{12}$$

Since $g_u(\lambda)$ is differentiable (see A.2) and $\mu_g-$strongly concave for $\lambda \in \mathcal{H}_\lambda$ :

$$g_u(\lambda) \leq g_u(\lambda_u^\star) + \nabla g_u(\lambda_u^\star)^T(\lambda - \lambda_u^\star) - \frac{\mu_g}{2}\|\lambda - \lambda_u^\star\|_2^2 \quad \forall \lambda \in \mathcal{H}_\lambda$$

From Lemma A.2 we have that $\nabla g_u(\lambda_u^\star) = \ell(\phi(\lambda_u^\star)))$, then evaluating at $\lambda_p^\star$ we obtain:

$$g_u(\lambda_p^\star) \leq g_u(\lambda_u^\star) + \ell(\phi(\lambda_u^\star))^T(\lambda_p^\star - \lambda_u^\star) - \frac{\mu_g}{2}\|\lambda_p^\star - \lambda_u^\star\|_2^2$$

By complementary slackness, $\ell(\phi(\lambda_u^\star))^T\lambda_u^\star = 0$. Then, since $\phi(\lambda_u^\star)$ is feasible and $\lambda_p^\star \geq 0$: $\ell(\phi(\lambda_u^\star))^T\lambda_p^\star \leq 0$. Thus,

$$g_u(\lambda_p^\star) \leq g_u(\lambda_u^\star) - \frac{\mu_g}{2}\|\lambda_p^\star - \lambda_u^\star\|_2^2$$

By Proposition 1: $g_p(\lambda_p^\star) - M\nu(1 + \|\lambda_p^\star\|_1) \leq g_u(\lambda_p^\star)$, which implies:

$$g_p(\lambda_p^\star) - M\nu(1 + \|\lambda_p^\star\|_1) \leq g_u(\lambda_u^\star) - \frac{\mu_g}{2}\|\lambda_p^\star - \lambda_u^\star\|_2^2$$

Thus,

$$\|\lambda_p^\star - \lambda_u^\star\|_2^2 \leq \frac{2}{\mu_g}\left[g_u(\lambda_u^\star) - g_p(\lambda_p^\star)\right] + \frac{2}{\mu_g}M\nu(1 + \|\lambda_p^\star\|_1) \tag{13}$$

Finally, since $\mathcal{F}_\theta \subseteq \mathcal{F}$ we have that : $g_u(\lambda) \leq g_p(\lambda) \quad \forall \lambda$. Evaluating at $\lambda_u^\star$ and using that $\lambda_p^\star$ maximizes $g_p$ we obtain:

$$g_u(\lambda_u^\star) \leq g_p(\lambda_u^\star)$$
$$\leq g_p(\lambda_p^\star)$$

Using this in equation 13 we obtain,

$$\|\lambda_p^\star - \lambda_u^\star\|_2^2 \leq \frac{2}{\mu_g}M\nu(1 + \|\lambda_p^\star\|_1)$$

## A.5 PROOF OF PROPOSITION 3.2: PERTURBATION OF DUAL VARIABLES

**Proposition.** *3.2 Under assumptions 3.1-3.4, the distance between the constraint violations of $\phi(\lambda_p^\star)$ and $\phi(\lambda_u^\star)$ is bounded by:*

$$\|\ell(\phi(\lambda_p^\star)) - \ell(\phi(\lambda_u^\star))\|_2^2 \leq 2\frac{\beta_g^2}{\mu_g}M\nu(1 + \|\lambda_p^\star\|_1) \tag{14}$$

The proof follows from straightforward applications of Lemma A.2 and Proposition A.3. Since the dual function $g_u(\lambda)$ is $\beta_g-$smooth, we have:

$$\|\ell(\phi(\lambda_p^\star)) - \ell(\phi(\lambda_u^\star))\|_2^2 = \|\nabla_\lambda g_u(\lambda_p^\star) - \nabla_\lambda g_u(\lambda_u^\star)\|_2^2$$
$$\leq \beta_g^2\|\lambda_p^\star - \lambda_u^\star\|_2^2$$

Then, the bound between optimal dual variables given in Proposition A.3 yields:

$$\|\ell(\phi(\lambda_p^\star)) - \ell(\phi(\lambda_u^\star))\|_2^2 \leq 2\frac{\beta_g^2}{\mu_g}M\nu(1 + \|\lambda_p^\star\|_1)$$

which concludes the proof.

## A.6 PROOF OF LEMMA 3.1: CURVATURE OF THE DUAL FUNCTION

**Lemma.** *3.1 Under assumptions 3.1, 3.2, 3.5 and 3.6, the unparametrized dual function $g_u(\lambda_u)$ is $\mu_g-$strongly concave and $\beta_g-$smooth on $\mathcal{H}_\lambda$ with:*

$$\mu_g = \frac{\mu_0\,\sigma^2}{\beta^2(1 + \Delta)^2} \quad and \quad \beta_g = \frac{\sqrt{m}M^2}{\mu_0} \tag{15}$$

*where $\Delta = \max(\|\lambda_u^\star\|_1, \|\lambda_p^\star\|_1)$.*

### A.6.1 STRONG CONCAVITY CONSTANT $\mu_g$

As shown in Lemma A.2, the unparametrized Lagrangian has a unique minimizer $\phi(\lambda)$ for each $\lambda \in \mathbb{R}_+^m$. Let $\lambda_1, \lambda_2 \in \mathcal{H}_\lambda$ and $\phi_1 = \phi(\lambda_1), \phi_2 = \phi(\lambda_2)$.

By convexity of the functions $\ell_i : \mathcal{F} \to \mathbb{R}$ for $i = 1, \ldots, m$, we have:

$$\ell_i(\phi_2) \geq \ell_i(\phi_1) + \langle D_\phi \ell_i(\phi_1), \phi_2 - \phi_1 \rangle,$$
$$\ell_i(\phi_1) \geq \ell_i(\phi_2) + \langle D_\phi \ell_i(\phi_2), \phi_1 - \phi_2 \rangle$$

Multiplying the above inequalities by $[\lambda_1]_i \geq 0$ and $[\lambda_2]_i \geq 0$ respectively and adding them, we obtain:

$$-\langle \ell(\phi_2) - \ell(\phi_1), \lambda_2 - \lambda_1 \rangle \geq \langle \lambda_1^T D_\phi \ell(\phi_1) - \lambda_2^T D_\phi \ell(\phi_2), \phi_2 - \phi_1 \rangle \tag{16}$$

Since $\nabla g_u(\lambda) = L(\phi(\lambda))$, we have that:

$$-\langle \nabla g_u(\lambda_2) - \nabla g_u(\lambda_2), \lambda_2 - \lambda_1 \rangle \geq \langle \lambda_1^T D_\phi L(\phi_1) - \lambda_2^T D_\phi L(\phi_2), \phi_2 - \phi_1 \rangle \tag{17}$$

Moreover, first order optimality conditions yield:

$$D_\phi \ell_0(\phi_1) + \lambda_1^T D_\phi \ell(\phi_1) = 0,$$
$$D_\phi \ell_0(\phi_2) + \lambda_2^T D_\phi \ell(\phi_2) = 0 \tag{18}$$

where 0 denotes the null-opereator from $\mathcal{F}$ to $\mathbb{R}$ (see e.g: (Kurdila & Zabarankin, 2006) Theorem 5.3.1).

Combining equations 17 and 18 we obtain:

$$-\langle \nabla g_u(\lambda_2) - \nabla g_u(\lambda_2), \lambda_2 - \lambda_1 \rangle \geq \langle D_\phi \ell_0(\phi_2) - D_\phi \ell_0(\phi_1), \phi_2 - \phi_1 \rangle$$
$$\geq \mu_0 \|\phi_2 - \phi_1\|_{L_2}^2 \tag{19}$$

where we used the $\mu_0-$strong convexity of the operator $\ell_0$.

We will now obtain a lower bound on $\|\phi_2 - \phi_1\|_{L_2}$, starting from the $\beta-$smoothness of $\ell_0$:

$$\|\phi_2 - \phi_1\|_2 \geq \frac{1}{\beta} \|D_\phi \ell_0(\phi_2) - D_\phi \ell_0(\phi_1)\|_{L_2}$$
$$= \frac{1}{\beta} \|\lambda_2^T D_\phi \ell(\phi_2) - \lambda_1^T D_\phi \ell(\phi_1)\|_{L_2} \tag{20}$$
$$= \frac{1}{\beta} \|(\lambda_2 - \lambda_1)^T D_\phi \ell(\phi_2) - \lambda_1^T (D_\phi \ell(\phi_1) - D_\phi \ell(\phi_2))\|_{L_2}$$

Then, second term in the previous equality can be characterized using assumption 3.6:

$$\|(\lambda_2 - \lambda_1)^T D_\phi \ell(\phi_2)\|_{L_2} \geq \sigma \|\lambda_2 - \lambda_1\|_2 \tag{21}$$

For the second term, using the $\beta-$smoothness of $\ell_i$ we can derive:

$$\|\lambda_1^T (D_\phi \ell(\phi_1) - D_\phi \ell(\phi_2))\|_{L_2} = \|\sum_{i=1}^{m} [\lambda_1]_i (D_\phi \ell_i(\phi_1) - D_\phi \ell_i(\phi_2))\|_{L_2}$$
$$\leq \sum_{i=1}^{m} [\lambda_1]_i \|D_\phi \ell_i(\phi_1) - D_\phi \ell_i(\phi_2)\|_{L_2} \tag{22}$$
$$\leq \sum_{i=1}^{m} [\lambda_1]_i \beta \|\phi_1 - \phi_2\|_{L_2}$$
$$= \beta \|\lambda_1\|_1 \|\phi_1 - \phi_2\|_{L_2}$$

Then, using the reverse triangle inequality:

$$\|(\lambda_2 - \lambda_1)^T D_\phi \ell(\phi_2) - \lambda_1^T (D_\phi \ell(\phi_1) - D_\phi \ell(\phi_2))\|_{L_2}$$
$$\geq \|(\lambda_2 - \lambda_1)^T D_\phi \ell(\phi_2)\|_{L_2} - \|\lambda_1^T (D_\phi \ell(\phi_1) - D_\phi \ell(\phi_2))\|_{L_2} \tag{23}$$
$$\geq \sigma \|\lambda_2 - \lambda_1\|_2 - \beta \|\lambda_1\|_1 \|\phi_2 - \phi_1\|_{L_2}$$

Combining this with equation 20 we obtain:

$$\|\phi_2 - \phi_1\|_2 \geq \frac{1}{\beta} \left( \sigma \|\lambda_2 - \lambda_1\|_2 - \beta \|\lambda_1\|_1 \|\phi_2 - \phi_1\|_{L_2} \right)$$
$$\longrightarrow \|\phi_2 - \phi_1\|_{L_2} \geq \frac{\sigma}{\beta(1 + \|\lambda_1\|_1)} \|\lambda_2 - \lambda_1\|_2 \tag{24}$$

This means that we can write equation 19 as:

$$-\langle \nabla g_u(\lambda_2) - \nabla g_u(\lambda_1), \lambda_2 - \lambda_1 \rangle \geq \frac{\mu_0 \sigma^2}{\beta^2 (1 + \|\lambda_1\|_1)^2} \|\lambda_2 - \lambda_1\|_2^2$$

Letting $\lambda_2 = \lambda_u^\star$, we obtain that the strong concavity constant of $g_u$ in $\mathcal{H}_\lambda$ is $\mu_g = \frac{\mu_0 \sigma^2}{\beta^2 (1 + \max\{\|\lambda_u^\star\|_1, \|\lambda_p^\star\|_1\})^2}$. A similar proof in the finite dimensional case can be found in (Guigues, 2020).

### A.6.2 SMOOTHNESS CONSTANT $\beta_g$

Set $\lambda_1, \lambda_2 \in \mathbb{R}_+^m$, and let $\phi_1 = \phi(\lambda_1)$ and $\phi_2 = \phi(\lambda_2)$ denote the Lagrangian minimizers associated to these multipliers.

Since the unparametrized Lagrangian is differentiable and $\mu_0$-strongly convex we have:

$$\mathcal{L}(f, \lambda) \geq \mathcal{L}(\phi(\lambda), \lambda) + \langle D_\phi \mathcal{L}(\phi(\lambda), \lambda)), f - \phi(\lambda)\rangle + \frac{\mu_0}{2}\|f - \phi(\lambda)\|_{L_2}^2$$

Using that $\phi(\lambda)$ is a minimizer, we obtain (see e.g: (Kurdila & Zabarankin, 2006) Theorem 5.3.1) :

$$\mathcal{L}(\phi(\lambda), \lambda) \leq \mathcal{L}(f, \lambda) - \frac{\mu_0}{2}\|f - \phi(\lambda)\|_2^2, \forall f \in \mathcal{F}$$

Applying this to $\phi_2$ and $\phi_1$ we obtain:

$$\ell_0(\phi_2) + \lambda_2^T \ell(\phi_2) \leq \ell_0(\phi_1) + \lambda_2^T \ell(\phi_1) - \frac{\mu_0}{2}\|\phi_2 - \phi_1\|_{L_2}^2$$

$$\ell_0(\phi_1) + \lambda_1^T \ell(\phi_1) \leq \ell_0(\phi_2) + \lambda_1^T \ell(\phi_2) - \frac{\mu_0}{2}\|\phi_2 - \phi_1\|_{L_2}^2$$

Summing the above inequalities and applying Cauchy-Schwarz:

$$\begin{aligned}
\mu_0\|\phi_2 - \phi_1\|_2^2 &\leq (\lambda_2 - \lambda_1)^T(\ell(\phi_1) - \ell(\phi_2)) \\
&\leq \|\lambda_2 - \lambda_1\|_2\|\ell(\phi_1) - \ell(\phi_2)\|_2 \\
&\leq \sqrt{m}M\|\lambda_2 - \lambda_1\|_2\|\phi_1 - \phi_2\|_{L_2}
\end{aligned}$$

where the last inequality follows from assumption 3.1. Then, applying Lemma A.2 we obtain:

$$\begin{aligned}
\|\nabla_\lambda g_u(\lambda_2) - \nabla_\lambda g_u(\lambda_1)\|_2 &= \|\ell(\phi_2) - \ell(\phi_1)\|_2 \\
&\leq M\|\phi_2 - \phi_1\|_{L_2} \\
&\leq \sqrt{m}\frac{M^2}{\mu_0}\|\lambda_2 - \lambda_1\|_2
\end{aligned}$$

which means that $g_u$ has a smoothness constant $\beta_g = \sqrt{m}\frac{M^2}{\mu_0}$.

## A.7 PROOF LEMMA A.4

**Lemma A.4.** *Let $P^\dagger$ denote the Fenchel conjugate of the perturbation function $P^\star(\epsilon)$. For every $\lambda \in \mathbb{R}_+^m$ we have that $P^\dagger(\lambda) = -g_u(\lambda)$.*

By definition of Fenchel conjugate:

$$P^\dagger(\lambda) = \sup_\epsilon \lambda^T \epsilon - P^\star(\epsilon) \tag{25}$$

Using the definition of the perturbation function $P^\star(\epsilon)$ we obtain:

$$\begin{aligned}
P^\dagger(\lambda) = \sup_{\phi \in \mathcal{F}, \epsilon} \quad & \lambda^T \epsilon - \ell_0(\phi) \\
\text{s.t: } & \ell(\phi) + \epsilon \preceq 0
\end{aligned} \tag{26}$$

Applying the change of variable $z = \ell(\phi) + \epsilon$, $P^\dagger(\lambda)$ can be written as:

$$\begin{aligned}
P^\dagger(\lambda) = \sup_{\phi \in \mathcal{F}, \mathbf{z}} \quad & \lambda^T \mathbf{z} - \lambda^T \ell(\phi) - \ell_0(\phi) \\
\text{s. to: } & \mathbf{z} \preceq 0
\end{aligned} \tag{27}$$

Since $\mathbf{z} \preceq 0$, the term $\lambda^T \mathbf{z}$ is unbounded above for $\lambda \prec 0$. Thus, we restrict the domain of $P^\dagger(\lambda)$ to $\lambda \succeq 0$. In this region, maximizing over $\mathbf{z} \in \mathbb{R}_-^m$ yields $\mathbf{z}^\star = 0$. We can thus write $P^\dagger(\lambda)$ as:

$$\begin{aligned}
P^\dagger(\lambda) &= \sup_{\phi \in \mathcal{F}} -\lambda^T \ell(\phi) - \ell_0(\phi), \quad \lambda \succeq 0 \\
&= -\inf_{\phi \in \mathcal{F}} \lambda^T \ell(\phi) + \ell_0(\phi), \quad \lambda \succeq 0
\end{aligned} \tag{28}$$

Therefore,

$$P^\dagger(\lambda) = -g_u(\lambda), \quad \lambda \succeq 0.$$

Similar versions of this result can be found in (Rockafellar, 1997), Section 28, (Guigues, 2020), Lemma 2.9 or (Rockafellar, 1974), Theorem 7.

## A.8 PROOF OF COROLLARY A.1: CURVATURE OF THE PERTURBATION FUNCTION

**Corollary A.1.** *Let $\mathcal{B}_\epsilon = \{\gamma\epsilon_{\mathbf{u}} + (1-\gamma)\epsilon_{\mathbf{p}} \; : \; \gamma \in [0,1]\}$ denote the segment connecting $\epsilon_{\mathbf{u}}$ and $\epsilon_{\mathbf{p}}$. The perturbation function $P^\star(\epsilon)$ is $\mu_{\mathbf{p}}$−strongly convex on $\mathcal{B}_\epsilon$ with constant: $\mu_\epsilon = 1/\beta_g$.*

We begin by stating a well-known Lemma on the duality between smoothness and strong convexity

**Lemma A.5.** *Let $h$ be a closed convex function defined on a subset of the vector space $\mathcal{X}$; $h$ is $\mu$−strongly convex if and only if $h^\dagger$ has $\mu$−Lipschitz continuous gradients. (See e.g, (Kakade et al., 2009) or (Goebel & Rockafellar, 2008)).*

In order to apply Lemma A.5 we need to show that the perturbation function $P(\epsilon)$ is convex and closed in the region of interest.

Convexity of $P^\star(\epsilon)$ for convex functional programs is shown in (Bonnans & Shapiro, 1998) or (Rockafellar, 1997) Theorem 29.1. Now we will show that $P^\star(\epsilon)$ is proper and lower semi continuous in the region of interest, which implies that it is closed.

The functional $\ell_0$, defined on the compact set $\mathcal{F}$, is smooth. Thus, it is bounded on $\mathcal{F}$. From assumption 3.2 we have that the problem is feasible for $\epsilon = 0$. Therefore, $P(0) < +\infty$. Moreover, by boundedness of $\ell_0$, $P(\epsilon) > -\infty \quad \forall \epsilon$, implying that $P(\epsilon)$ is proper.

Now, fix $\epsilon^0 \in \mathcal{B}_\epsilon$. Assumption 3.2 implies that the perturbed problem with constraint: $L(f)+\epsilon^0 \preceq 0$ is strictly feasible. Since this perturbed problem is convex and strictly feasible, its perturbation function $\tilde{P}(\epsilon)$ is lower semi continuous at $0$ (see (Bonnans & Shapiro, 1998) Theorem 4.2). Note that $\tilde{P}(\epsilon) = P^\star(\epsilon + \epsilon^0)$. Thus, $P^\star(\epsilon)$ is lower semi continuous at $\epsilon^0$.

We conclude that $P^\star(\epsilon)$ is proper and lower semi continuous for all $\epsilon \in \mathcal{B}_\epsilon$.

On the other hand, from Corollary 3.1 $P^\dagger(\lambda) = -g_u(\lambda)$ is $\beta_g$-smooth on $\mathbb{R}_+^m$. Thus, we are in the hypothesis of Proposition A.4, which implies that $P^\star(\epsilon)$ is strongly convex on $\mathcal{B}_\epsilon$ with constant $\frac{1}{\beta_g}$.

## A.9 PROOF OF PROPOSITION A.1: SUBGRADIENTS OF $P^\star$

**Proposition A.1.** *Under assumptions 3.1 and 3.2, $\lambda_p^\star$ is a subgradient of the perturbation function at $\epsilon_u$. That is, $\lambda_p^\star \in \partial P^\star(\epsilon_u)$.*

The conjugate nature of the dual function $g_u(\lambda)$ and the perturbation function $P^\star(\epsilon)$ also establishes a dependence between their first order variations. This dependence is captured in the following lemma.

**Lemma A.6.** *If $h$ is a closed convex function, the subdifferential $\partial h^\dagger$ is the inverse of $\partial h$ in the sense of multivalued mappings (see (Rockafellar, 1997) Corollary 23.5.1):*

$$x \in \partial h^\dagger(y) \iff y \in \partial h(x)$$

On one hand, from Lemma A.2, we have that $\nabla_\lambda g_u(\lambda_p^\star) = \ell(\phi(\lambda_p^\star)) = -\epsilon_u$. On the other hand, from Lemma A.4, $P^\dagger(\lambda) = -g_u(\lambda)$ for all $\lambda \in \mathbb{R}_+^m$.

Taking the gradient with respect to $\lambda$ and evaluating at $\lambda_p^\star$ we obtain: $\nabla_\lambda P^\dagger(\lambda_p^\star) = \epsilon_u$. Then, Lemma A.6, yields the sensitivity result:

$$\lambda_p^\star \in \partial P^\star(\epsilon_u).$$

## A.10 PROOF OF PROPOSITION A.2: DISTANCE BETWEEN OPTIMAL VALUES

**Proposition A.2.** *Under assumptions 3.3 and 3.1, the difference between the optimal values of problems perturbed by $\epsilon_p$ and $\epsilon_u$ is bounded:*

$$P^\star(\epsilon_p) - P^\star(\epsilon_u) \leq M\nu(1 + \|\lambda_p^\star\|) + \lambda_p^{*T}(\epsilon_p - \epsilon_u)$$

Recall that $\epsilon_u = -\ell(\phi(\lambda_p^\star))$ and $\epsilon_p = -\ell(f_\theta(\lambda_p^\star))$. We want to show that:

$$P^\star(\epsilon_p) - P^\star(\epsilon_u) \leq M\nu(1 + \|\lambda_p^\star\|) + \lambda_p^{*T}(\epsilon_p - \epsilon_u)$$

We start by showing that $P^\star(\epsilon_p) \leq \ell_0(f_\theta(\lambda_p^\star))$. Note that $f_\theta(\lambda_p^\star)$ is feasible in the perturbed problem, since its constraint value is $-\epsilon_p$. Then,

$$P(\epsilon_p) = \min_f \{L_0(f) \,:\, L(f) + \epsilon_p \preceq 0\} \leq L_0(f_\theta(\lambda_p^\star))$$

Therefore,

$$P^\star(\epsilon_p) - P^\star(\epsilon_u) \leq \ell_0(f_\theta(\lambda_p^\star)) - P^\star(\epsilon_u). \tag{29}$$

Note that the dual function of the problem perturbed by $\epsilon_u$ is $\tilde{g}_u(\lambda, \epsilon_u) := \min_{\phi \in \mathcal{F}} \{\ell_0(f) + \lambda^T(\ell(\phi) + \epsilon_u)\}$. Then, weak duality implies that $P^\star(\epsilon_u) \geq \tilde{g}_u(\lambda, \epsilon_u)$ for all $\lambda$. Evaluating at $\lambda_p^\star$ we obtain:

$$\begin{aligned}
P^\star(\epsilon_u) &\geq \min_{\phi \in \mathcal{F}} \left\{ L_0(f) + \lambda_p^{*T}(L(f) + \epsilon_u) \right\} \\
&= \min_{\phi \in \mathcal{F}} \left\{ \ell_0(\phi) + \lambda_p^{*T}\ell(\phi) \right\} + \lambda_p^{*T}\epsilon_u \\
&= g_u(\lambda_p^\star) + \lambda_p^{*T}\epsilon_u
\end{aligned} \tag{30}$$

Combining equations 30 and 29 we obtain:

$$\begin{aligned}
P^\star(\epsilon_p) - P^\star(\epsilon_u) &\leq \ell_0(f_\theta(\lambda_p^\star)) - g_u(\lambda_p^\star) - \lambda_p^{*T}\epsilon_u \\
&= \ell_0(f_\theta(\lambda_p^\star)) \pm \lambda_p^{*T}\epsilon_p - g_u(\lambda_p^\star) - \lambda_p^{*T}\epsilon_u
\end{aligned} \tag{31}$$

Recall that $\epsilon_p = -\ell(f_\theta(\lambda_p^\star))$. Then, we can identify the parametrized dual function $g_p(\lambda) = \ell_0(f_\theta(\lambda_p^\star)) - \lambda_p^{*T}\epsilon_p$ and write equation 31 as:

$$P^\star(\epsilon_p) - P^\star(\epsilon_u) \leq g_p(\lambda_p^\star) - g_u(\lambda_p^\star) + \lambda_p^{*T}(\epsilon_p - \epsilon_u)$$

Finally, leveraging the bound between dual functions from Lemma A.1 we obtain:

$$P^\star(\epsilon_p) - P^\star(\epsilon_u) \leq M\nu(1 + \|\lambda_p^\star\|_1) + \lambda_p^{*T}(\epsilon_p - \epsilon_u),$$

which conludes the proof.

### A.11 PROOF OF PROPOSITION 3.3: FUNCTION CLASS PERTURBATION

**Proposition.** *3.3 Under assumptions 3.1-3.4, the distance between constraint violation associated to the parametrization of the hypothesis class is given by:*

$$\|\ell(\phi(\lambda_p^\star)) - \ell(f_\theta(\lambda_p^\star))\|_2^2 \leq 2\beta_g M\nu(1 + \|\lambda_p^\star\|_1)$$

Let $\Delta\epsilon = \epsilon_p - \epsilon_u$, using the strong convexity constant obtained in Proposition A.1 we have that:

$$P^\star(\epsilon_p) \geq P^\star(\epsilon_u) + s^T\Delta\epsilon + \frac{1}{2\beta_g}\|\Delta\epsilon\|_2^2$$

where $s \in \partial P^\star(\epsilon_u)$ is a subgradient of $P^\star(\epsilon)$ at $\epsilon_u$.

From Proposition A.1 we know that: $\lambda_p^\star \in \partial P^\star(\epsilon_u)$. Thus,

$$P^\star(\epsilon_p) \geq P^\star(\epsilon_u) + \lambda_p^{*T}\Delta\epsilon + \frac{1}{2\beta_g}\|\Delta\epsilon\|_2^2$$

Using the bound on $P^\star(\epsilon_p) - P^\star(\epsilon_u)$ obtained in proposition A.2 we can write:

$$M\nu(1 + \|\lambda_p^\star\|_1) + \lambda_p^{*T}\Delta\epsilon \geq \lambda_p^{*T}\Delta\epsilon + \frac{1}{2\beta_g}\|\Delta\epsilon\|_2^2$$

$$\longrightarrow M\nu(1 + \|\lambda_p^\star\|_1) \geq \frac{1}{2\beta_g}\|\Delta\epsilon\|_2^2$$

This implies:

$$\|\Delta\epsilon\|_2^2 \leq 2\beta_g M\nu(1 + \|\lambda_p^\star\|_1)$$

$$\longrightarrow \|\ell(\phi) - \ell(f_\theta(\lambda_p^\star))\|_2^2 \leq 2\beta_g M\nu(1 + \|\lambda_p^\star\|_1)$$

which concludes the proof.

## A.12 PROOF OF PROPOSITION 3.1: 2-NORM NEAR-FEASIBILITY

**Proposition. 3.1** *Under assumptions 3.1-3.4, for all $f_\theta(\lambda_p^\star) \in \mathcal{F}_\theta^\star(\lambda_p^\star)$, the distance between the unparametrized and parametrized constraint violations is bounded by:*

$$\|\ell(f_\theta(\lambda_p^\star)) - \ell(\phi^\star)\|_2^2 \leq 2\beta_g M\nu(1 + \|\lambda_p^\star\|_1)\left(1 + \sqrt{\frac{\beta_g}{\mu_g}}\right)^2$$

Proposition 3.1 stems from combining the feasibility bounds in Corollary 3.2 and Proposition 3.3:

$$\|\ell(\phi(\lambda_p^\star)) - \ell(\phi^\star)\|_2 \leq \sqrt{2\frac{\beta_g^2}{\mu_g}M\nu(1 + \|\lambda_p^\star\|_1)} \tag{32}$$

$$\|\ell(\phi(\lambda_p^\star)) - \ell(f_\theta(\lambda_p^\star))\|_2 \leq \sqrt{2\beta_g M\nu(1 + \|\lambda_p^\star\|_1)} \tag{33}$$

Combining the above equations through a triangle inequality we obtain:

$$\|\ell(f_\theta(\lambda_p^\star)) - \ell(\phi(\lambda_u^\star))\|_2 \leq \sqrt{2\frac{\beta_g^2}{\mu_g}M\nu(1 + \|\lambda_p^\star\|_1)} + \sqrt{2\beta_g M\nu(1 + \|\lambda_p^\star\|_1)} \tag{34}$$

$$= \sqrt{2\beta_g M\nu(1 + \|\lambda_p^\star\|_1)}\left(1 + \sqrt{\frac{\beta_g}{\mu_g}}\right) \tag{35}$$

Taking squares on both sides yields the desired result.

## A.13 PROOF OF THEOREM 3.1: NEAR-FEASIBILITY AND NEAR-OPTIMALITY

**Theorem. 3.1** *Under assumptions 3.1, 3.2, 3.3, 3.5 and 3.6, the sub-optimality and near-feasibility of all $f_\theta(\lambda_p^\star) \in \mathcal{F}(\lambda_p^\star)$ is bounded by*

$$\|\ell(f_\theta(\lambda_p^\star)) - \ell(\phi^\star)\|_\infty \leq M\left[1 + \kappa_1\kappa_0(1 + \Delta)\right]\sqrt{2m\frac{M\nu}{\mu_0}(1 + \|\lambda_p^\star\|_1)} := \Gamma_2 \tag{36}$$

$$|P_p^\star - \ell_0(f_\theta(\lambda_p^\star))| \leq \|\lambda_p^\star\|_1\Gamma_2 + (1 + \|\lambda_p^\star\|_1)M\nu + \Gamma_1 \tag{37}$$

*with $\kappa_1 = \frac{M}{\sigma}$, $\kappa_0 = \frac{\beta}{\mu_0}$, $\Delta = \max\{\|\lambda_u^\star\|_1, \|\lambda_p^\star\|_1\}$ and $\Gamma_1$ as in eq. 4.*

### A.13.1 NEAR-FEASIBILITY

Recall that Lemma 3.1 characterizes the strong concavity $\mu_g$ and smoothness $\beta_g$ of the dual function in terms of the properties of the losses $\ell_i$ and the functional space $\mathcal{F}$. The proof of this theorem stems from applying Lemma 3.1 to the 2-norm bound in Theorem 3.1.

We start by observing that:

$$\|\ell(f_\theta(\lambda_p^\star)) - \ell(\phi(\lambda_u^\star))\|_\infty \leq \|\ell(f_\theta(\lambda_p^\star)) - \ell(\phi(\lambda_u^\star))\|_2 \tag{38}$$

$$\leq \sqrt{2\beta_g M\nu(1 + \|\lambda_p^\star\|_1)}\left(1 + \sqrt{\frac{\beta_g}{\mu_g}}\right) \tag{39}$$

From proposition 3.1, we have that $\mu_g = \frac{\mu_0\,\sigma^2}{\beta^2(1+\Delta)^2}$ and $\beta_g = \frac{\sqrt{m}M^2}{\mu_0}$. This implies that

$$\frac{\beta_g}{\mu_g} = \sqrt{m}\frac{M^2}{\sigma^2}\frac{\beta^2}{\mu_0^2}(1 + \Delta)^2$$

where $\Delta = \max\{\|\lambda_u^\star\|_1, \|\lambda_p^\star\|_1\}$. Plugging this into equation 39, we obtain:

$$\|\ell(f_\theta(\lambda_p^\star)) - \ell(\phi(\lambda_u^\star))\|_\infty \leq Mm^{1/4}\sqrt{2\frac{M\nu}{\mu_0}(1 + \|\lambda_p^\star\|_1)}\left[1 + m^{1/4}\frac{M}{\sigma}\frac{\beta}{\mu_0}(1 + \Delta)\right]$$

$$\leq M\sqrt{2\frac{M\nu}{\mu_0}(1 + \|\lambda_p^\star\|_1)}\left[1 + \frac{M}{\sigma}\frac{\beta}{\mu_0}(1 + \Delta)\right]\sqrt{m}$$

Finally, using the definitions of the condition numbers $\kappa_1 = \frac{M}{\sigma}$, $\kappa_0 = \frac{\beta}{\mu_0}$ we obtain:

$$\|\ell(f_\theta(\lambda_p^\star)) - \ell(\phi(\lambda_u^\star))\|_\infty \leq M \left[1 + \kappa_1 \kappa_0 (1 + \Delta)\right] \sqrt{2m \frac{M\nu}{\mu_0}(1 + \|\lambda_p^\star\|_1)} \tag{40}$$

which is the desired near-feasibility bound.

### A.13.2 NEAR-OPTIMALITY

To derive the near-optimality bound, we combine equation 40 with the duality gap bound from (Chamon et al., 2023, Prop. 3.3):

$$P_p^\star - D_p^\star \leq M\nu(1 + \|\tilde{\lambda}^\star\|_1) := \Gamma_1, \tag{41}$$

where $\tilde{\lambda}^\star$ maximizes $\tilde{g}_p(\lambda) = g_p(\lambda) + M\nu\|\lambda\|_1$.

Since $P_p^\star \geq P_u^\star$ we have:

$$P_u^\star - D_p^\star \leq \Gamma_1 \Leftrightarrow \ell_0(\phi^*) - \ell_0(f_\theta(\lambda_p^*)) \leq \lambda_p^{*T}\ell(f_\theta(\lambda_p^*)) + \Gamma_1$$

Then, using that the solution of the unparametrized problem $\phi^*$ is feasible (i.e, $\ell(\phi^*)) \leq 0$) and $\lambda_p^* \succeq 0$ we obtain

$$\ell_0(\phi^*) - \ell_0(f_\theta(\lambda_p^*)) \leq \lambda_p^{*T}(\ell(f_\theta(\lambda_p^*)) - \ell(\phi^*)) + \Gamma_1$$
$$\leq \|\lambda_p^*\|_1 \|\ell(\phi^*) - \ell(f_\theta(\lambda_p^*))\|_\infty + \Gamma_1$$

To conclude the derivation, note that the $\nu$-universality and $M$-Lipschitz continuity (Assumptions 3.3 and 3.1) imply that there exists $\theta'$ such that $|\ell_i(\phi^*) - \ell_i(f_{\theta'})| \leq M\nu$ for all $i = 0, \ldots, m$. Thus,

$$\ell_0(\phi^*) - \ell_0(f_\theta(\lambda_p^*)) \geq \ell_0(\phi^*) \pm \ell_0(f_{\theta'}) - \ell_0(f_\theta(\lambda_p^*))$$
$$\geq \ell_0(f_{\theta'}) - M\nu - \ell_0(f_\theta(\lambda_p^*)) \tag{42}$$

Note that the duality gap bound implies the approximate saddle-point relation:

$$L(f_\theta(\lambda_p^*), \lambda_u^*) - \Gamma_1 \leq L(f_\theta(\lambda_p^*), \lambda_p^*) \leq L(f_{\theta'}, \lambda_p^*) + \Gamma_1.$$

Applying the right-hand side of this inequality to equation 42 we obtain:

$$\ell_0(\phi^*) - \ell_0(f_\theta(\lambda_p^*)) \geq \lambda_p^{*T}(\ell(f_\theta(\lambda_p^*)) - \ell(\phi^*)) - \Gamma_1 - (1 + \|\lambda_p^*\|_1)M\nu$$
$$\geq -\|\lambda_p^*\|_1 \|\ell(\phi^*) - \ell(f_\theta(\lambda_p^*))\|_\infty - (1 + \|\lambda_p^*\|_1)M\nu - \Gamma_1$$

which completes the proof.

### A.14 PROOF OF LEMMA 4.1: BEST ITERATE CONVERGENCE

**Lemma.** *4.1 Let $g_p^{best}(t|\lambda(t_0)) = \max_{s \in [t_0, t]} g_p(\lambda(s))$ be the maximum value of the parametrized dual function up to time $t$. Then,*

$$\lim_{t \to \infty} g_p^{best}(t|\lambda(t_0)) \geq D_p^\star - \left(\frac{\eta S^2}{2} + \rho\right) \quad a.s.$$

*where $S^2 > \mathbb{E}[\|\hat{s}(t)\|^2|\lambda(t)]$ is an upper bound on the norm of the second order moment of the stochastic supergradients.*

A similar proof in the context of resource allocation for wireless communications can be found in (Ribeiro, 2010), Theorem 2. To ease the notation, we will denote the value of the parametrized dual function at iteration $t$ by $g(t) := g_p(\lambda(t))$. Similarly, $g^{best}(t)$ will denote the largest value of $g(t)$ encountered so far.

We start by deriving a recursive inequality between the distances of iterates $\lambda(t)$ and an optimal dual variable $\lambda_p^\star \in \arg\max_{\lambda \succeq 0} g_p(\lambda)$.

**Proposition A.3.** *Consider the dual ascent algorithm described in Section 4 using a constant step size $\eta > 0$. Then,*

$$\mathbb{E}\{\|\lambda(t+1) - \lambda_p^\star\|^2 | \lambda(t)\} \leq \|\lambda(t) - \lambda_p^\star\|^2 + \eta^2 S^2 - 2\eta(D_p^\star - g(t) - \rho) \tag{43}$$

We delay the proof of Proposition A.3 to section A.14.1. We can observe that as the optimality gap $D_p^\star - g(t)$ decreases, the fixed term $\eta^2 S^2$ dominates the right hand side of equation (43), suggesting convergence of $\lambda(t)$ only to a neighborhood of $\lambda_p^\star$. In order to show this, the main obstacle is that Proposition A.3 bounds the expected value of $\|\lambda(t+1) - \lambda_p^\star\|^2$ and we wish to establish almost sure convergence. This can be addressed by leveraging the Supermartingale Convergence Theorem (see e.g, (Solo & Kong, 1994) Theorem E7.4), which we state here for completeness.

**Theorem A.1.** *Consider nonnegative stochastic processes $A(\mathbb{N})$ and $B(\mathbb{N})$ with realizations $\alpha(\mathbb{N})$ and $\beta(\mathbb{N})$ having values $\alpha(t) \geq 0$ and $\beta(t) \geq 0$ and a sequence of nested $\sigma$-algebras $\mathcal{A}(0:t)$ measuring at least $\alpha(0:t)$ and $\beta(0:t)$. If*

$$\mathbb{E}[\alpha(t+1) \mid \mathcal{A}(0:t)] \leq \alpha(t) - \beta(t) \tag{44}$$

*the sequence $\alpha(t)$ converges almost surely and $\beta(t)$ is almost surely summable, i.e., $\sum_{u=1}^\infty \beta(u) < \infty$ a.s.*

We define $\alpha(t)$ and $\beta(t)$ as follows,

$$\alpha(t) := \|\lambda(t) - \lambda_p^\star\|^2 \, \mathbb{I}\left\{ D_p^\star - g^{\text{best}}(t) > \frac{\eta S^2}{2} + \rho \right\}$$

$$\beta(t) := [2\eta(D_p^\star - g(t) - \rho) - \eta^2 S^2] \, \mathbb{I}\left\{ D_p^\star - g^{\text{best}}(t) > \frac{\eta S^2}{2} + \rho \right\}$$

Note that $\alpha(t)$ tracks $\|\lambda(t) - \lambda_p^\star\|^2$ until the optimality gap $D_p^\star - g^{\text{best}}(t)$ falls bellow the threshold $\frac{\eta S^2}{2} + \rho$ and is then set to 0. Similarly, $\beta(t)$ tracks $2\eta(D_p^\star - g(t) - \rho) - \eta^2 S^2$ until the optimality gap $D_p^\star - g^{\text{best}}(t)$ falls bellow the same threshold and is then set to 0.

It is clear that $\alpha(t) \geq 0$, since it is the product of a norm and an indicator function. The same holds for $\beta(t)$, since the indicator evaluates to 0 whenever $2\eta(D_p^\star - g(t) - \rho) - \eta^2 S^2 \leq 0$. We thus have, $\alpha(t), \beta(t) \geq 0$ for all $t$.

We will leverage Theorem A.1 to show that $\beta(t)$ is almost surely summable. Let $\mathcal{A}(0:t)$ be a sequence of $\sigma$-algebras measuring $\alpha(0:t), \beta(0:t)$ and $\lambda(0:t)$. We will show that $\alpha(t)$ and $\beta(t)$ satisfy the hypothesis of Theorem A.1 with respect to $\mathcal{A}(0:t)$. Note that at each iteration, $\alpha(t)$ and $\beta(t)$ are fully determined by $\lambda(t)$. Therefore, conditioning on $\mathcal{A}(0:t)$ is equivalent to conditioning on $\lambda(t)$, i.e: $\mathbb{E}\{\alpha(t)|\mathcal{A}(0:t)\} = \mathbb{E}\{\alpha(t)|\lambda(t)\}$. Then we can write,

$$\begin{aligned}
\mathbb{E}\{\alpha(t)|\mathcal{A}(0:t)\} =&\mathbb{E}\{\alpha(t)|\lambda(t), \alpha(t) = 0\}\mathbb{P}\{\alpha(t) = 0\} \\
&+ \mathbb{E}\{\alpha(t)|\lambda(t), \alpha(t) > 0\}\mathbb{P}\{\alpha(t) > 0\}
\end{aligned} \tag{45}$$

From equation 45, we will derive that $\mathbb{E}\{\alpha(t)|\mathcal{A}(0:t)\} \leq \alpha(t) - \beta(t)$ which is the remaining hypothesis in Theorem A.1.

On one hand, observe that if $\alpha(t) = 0$ we have that $\mathbb{I}\{D_p^\star - g^{\text{best}}(t) \leq \frac{\eta S^2}{2} + \rho\} = 0$. This is because in the case where $\|\lambda(t) - \lambda_p^\star\|^2 = 0$, the indicator function also evaluates to 0. Therefore, if $\alpha(t) = 0$, it must be that $\beta(t) = 0$. Then, trivially, $\mathbb{E}\{\alpha(t)|\lambda(t), \alpha(t) = 0\} = \alpha(t) - \beta(t)$.

On the other hand, when $\alpha(t) > 0$:

$$\mathbb{E}[\alpha(t+1) \mid \lambda(t), \alpha(t) > 0] \tag{46}$$

$$= \mathbb{E}\left\{ \|\lambda(t+1) - \lambda_p^\star\|^2 \, \mathbb{I}\left\{ D_p^\star - g^{\text{best}}(t+1) > \frac{\eta \hat{S}^2}{2} + \rho \right\} \mid \lambda(t) \right\} \tag{47}$$

$$\leq \mathbb{E}\left\{ \|\lambda(t+1) - \lambda_p^\star\|^2 \mid \lambda(t) \right\} \tag{48}$$

where we used the definition of $\alpha(t+1)$ and the fact the the indicator function is not larger than 1. Then, from proposition A.3 we have:

$$\mathbb{E}[\alpha(t+1) \mid \lambda(t), \alpha(t) > 0] \leq \left\| \lambda(t) - \lambda_p^\star \right\|^2 + \eta^2 S^2 - 2\eta(D_p^\star - g(t) - \rho) \tag{49}$$
$$= \alpha(t) - \beta(t). \tag{50}$$

where the last equality comes from the fact that $\alpha(t) > 0$ implies $\mathbb{I}\left\{ D_p^\star - g^{\text{best}}(t+1) > \frac{\eta\hat{S}^2}{2} + \rho \right\} = 1$.

This means that we can write equation 45 as:

$$\mathbb{E}\{\alpha(t)|\mathcal{A}(0:t)\} \leq [\alpha(t) - \beta(t)](\mathbb{P}\{\alpha(t) = 0\} + \mathbb{P}\{\alpha(t) > 0\})$$
$$= \alpha(t) - \beta(t) \tag{51}$$

which shows that $\alpha(t)$ and $\beta(t)$ satisfy the hypothesis of Theorem A.1. Then, we have that $\beta(t)$ is almost surely summable, which implies,

$$\liminf_{t \to \infty} \left[ 2\eta(D_p^\star - g(t) - \rho) - \eta^2 S^2 \right] \mathbb{I}\{D_p^\star - g^{\text{best}}(t) > \eta\hat{S}^2 + \rho/2\} = 0 \text{ a.s.}$$

This is true if either $D_p^\star - g^{\text{best}}(t) \leq \frac{\eta S^2}{2} + \rho$ for some $t$, or if $\liminf_{t \to \infty} \left[ 2\eta(D_p^\star - g(t) - \rho) - \eta^2 S^2 \right] = 0$, which concludes the proof.

### A.14.1  PROOF OF PROPOSITION A.3

**Proposition.** *A.3 Consider the stochastic supergradient ascent algorithm from Section 4 using a constant step size $\eta > 0$. Then,*

$$\mathbb{E}\{\|\lambda(t+1) - \lambda_p^\star\|^2 | \lambda(t)\} \leq \|\lambda(t) - \lambda_p^\star\|^2 + \eta^2 S^2 - 2\eta(D_p^\star - g(t) - \rho) \tag{52}$$

Let $\hat{s}(t)$ denote the approximate stochastic supergradient $\hat{\ell}_i(f_\theta^\dagger(t))$. From the definition of $\lambda(t+1)$:

$$\|\lambda(t+1) - \lambda_p^\star\|^2 = \|[\lambda(t) + \eta\hat{s}(t)]_+ - \lambda_p^\star\|^2$$
$$\leq \|\lambda(t) - \lambda_p^\star + \eta\hat{s}(t)\|^2 \tag{53}$$
$$= \|\lambda(t) - \lambda_p^\star\|^2 + \eta^2\|\hat{s}(t)\|^2 + 2\eta\hat{s}(t)^T(\lambda(t) - \lambda_p^\star)$$

where we used the fact that setting the negative components of $\lambda(t) + \eta\hat{s}(t)$ to 0 decreases its distance to the positive vector $\lambda_p^\star$ and then expanded the square.

Note that for a given $\lambda(t)$, the relations in 53 hold for all realizations of $\hat{s}(t)$. Thus, the expectation of $\|\lambda(t+1) - \lambda_p^\star\|$, conditioned on $\lambda(t)$ satisifes:

$$\mathbb{E}\{\|\lambda(t+1) - \lambda_p^\star\|^2 | \lambda(t)\} \leq \|\lambda(t) - \lambda_p^\star\|^2 + \eta^2\mathbb{E}\{\|\hat{s}(t)\|^2 | \lambda(t)\} + 2\eta\mathbb{E}\{\hat{s}(t)|\lambda(t)\}(\lambda(t) - \lambda_p^\star) \tag{54}$$

Furthermore, the stochastic supergadient $\hat{s}(t)$ yields, on average, an approximate ascent direction of the dual function $g_p$:

$$\mathbb{E}\{\hat{s}(t)|\lambda(t)\}(\lambda(t) - \lambda) - \rho \leq g(t) - g_p(\lambda). \tag{55}$$

Evaluating the previous inequality at $\lambda_p^\star$ and combining it with equation 54 we obtain:

$$\mathbb{E}\{\|\lambda(t+1) - \lambda_p^\star\|^2 | \lambda(t)\} \leq \|\lambda(t) - \lambda_p^\star\|^2 + \eta^2 S^2 + 2\eta(g(t) - D_p^\star + \rho) \tag{56}$$

which concludes the proof.

### A.15  PROOF PROPOSITION 4.1

We will bound the distance between $\ell(\phi(\lambda_u^\star))$ and $\ell(f_\theta(\lambda^{\text{best}}))$ by partioning it into terms that we have previously analyzed in Corollary 3.2 and Proposition 3.3:

$$\|\ell(\phi(\lambda_u^\star)) - \ell(f_\theta(\lambda^{\text{best}}))\|_2 \leq \|\ell(\phi(\lambda_u^\star)) - \ell(\phi(\lambda^{\text{best}}))\|_2$$
$$+ \|\ell(\phi(\lambda^{\text{best}})) - \ell(f_\theta(\lambda^{\text{best}}))\|_2$$

The first term is of the same nature as the one analyzed in Corollary 3.2, since it is characterizes a perturbation in dual variables in the unparametrized problem. Thus, using the characterization of the curvature of the dual function from proposition A.3 and the sub-optimality of $\lambda^{\text{best}}$ with respect to $\lambda_p^\star$, this term can be bounded.

We denote by $\mathcal{B}_{\lambda^{\text{best}}}$ the segment connecting $\lambda^{\text{best}}$ and $\lambda_u^\star$ and by $\tilde{\mu}_g$ the strong concavity constant of $g_u$ in $\mathcal{B}_{\lambda^{\text{best}}}$. Using Lemma A.1 and the fact that $g_p(\lambda_p^\star) \geq g_u(\lambda_u^\star)$ we obtain:

$$\|\lambda^{\text{best}} - \lambda_u^\star\|_2^2 \leq \frac{2}{\tilde{\mu}_g}(g_u(\lambda_u^\star) - g_u(\lambda^{\text{best}}))$$

$$\leq \frac{2}{\tilde{\mu}_g}(g_p(\lambda_p^\star) - (g_p(\lambda^{\text{best}}) - M\nu(1 + \|\lambda^{\text{best}}\|_1)))$$

Then, leveraging the almost sure convergence shown in Proposition 4.1 we have:

$$\|\lambda^{\text{best}} - \lambda_u^\star\|_2^2 \leq \frac{2}{\tilde{\mu}_g}\left(M\nu(1 + \|\lambda^{\text{best}}\|_1) + \frac{\eta S^2}{2} + \rho\right) \tag{57}$$

Note that equation 57 corresponds to the bound in Proposition A.4 but amplified by the sub-optimality of $\lambda^{\text{best}}$ with respect to $\lambda_p^\star$. Then, since the gradient $\nabla_\lambda g_u(\lambda) = \ell(\phi(\lambda))$ is $\beta_g$-Lipschitz continuous:

$$\|\ell(\phi(\lambda^{\text{best}})) - \ell(\phi(\lambda_u^\star))\|_2^2 = \|\nabla_\lambda g_u(\lambda^{\text{best}}) - \nabla_\lambda g_u(\lambda_u^\star)\|_2^2 \tag{58}$$

$$\leq \beta_g^2 \|\lambda^{\text{best}} - \lambda_u^\star\|_2^2 \tag{59}$$

$$\leq \frac{2\beta_g^2}{\tilde{\mu}_g}\left(M\nu(1 + \|\lambda^{\text{best}}\|_1) + \frac{\eta S^2}{2} + \rho\right) \tag{60}$$

which completes the first part of the proof.

The term $\|\ell(\phi(\lambda^{\text{best}})) - \ell(f_\theta(\lambda^{\text{best}}))\|_2$ captures a perturbation in the function class for a fixed dual variable, and can be bounded by leveraging the perturbation analysis of Proposition 3.3. Let $\tilde{\epsilon}_u = -\ell(\phi(\lambda^{\text{best}}))$ and $\tilde{\epsilon}_p = -\ell(f_\theta(\lambda^{\text{best}}))$. First note that the duality between smoothness and strong convexity detailed in Corollary A.1 implies that $P^\star(\epsilon)$ is strongly convex with constant $\frac{1}{\beta_g}$ on $\mathcal{B}_{\lambda^{\text{best}}}$. Then, as in Proposition A.2, we can bound the distance between the optimal values associated to these perturbations $\tilde{\epsilon}_p$ and $\tilde{\epsilon}_u$ by:

$$P^\star(\tilde{\epsilon}_p) - P^\star(\tilde{\epsilon}_u) \leq M\nu(1 + \|\lambda^{\text{best}}\|_1) + \lambda^{{\text{best}}^T}(\tilde{\epsilon}_p - \tilde{\epsilon}_u) \tag{61}$$

The strong convexity of $P^\star$ combined with equation 61 yields:

$$M\nu(1 + \|\lambda^{\text{best}}\|_1) + \lambda^{{\text{best}}^T}\Delta\tilde{\epsilon} \geq \lambda^{{\text{best}}^T}\Delta\tilde{\epsilon} + \frac{1}{2\beta_g}\|\Delta\tilde{\epsilon}\|_2^2 \tag{62}$$

which implies:

$$\|\Delta\tilde{\epsilon}\|_2^2 \leq 2\beta_g M\nu(1 + \|\lambda^{\text{best}}\|_1) \tag{63}$$

We conclude the proof by summing the bounds in equations 60 and 63 to obtain:

$$\|\ell(\phi(\lambda_u^\star)) - \ell(f_\theta(\lambda^{\text{best}})))\|_2 \tag{64}$$

$$\leq \sqrt{2\beta_g M\nu(1 + \|\lambda^{\text{best}}\|_1)} + \sqrt{\frac{2\beta_g^2}{\tilde{\mu}_g}\left(M\nu(1 + \|\lambda^{\text{best}}\|_1) + \frac{\eta S^2}{2} + \rho\right)} \tag{65}$$

$$\leq \sqrt{2\beta_g\left(M\nu(1 + \|\lambda^{\text{best}}\|_1) + \frac{\eta S^2}{2} + \rho\right)} + \sqrt{\frac{2\beta_g^2}{\tilde{\mu}_g}\left(M\nu(1 + \|\lambda^{\text{best}}\|_1) + \frac{\eta S^2}{2} + \rho\right)} \tag{66}$$

$$= \sqrt{2\beta_g\left(M\nu(1 + \|\lambda^{\text{best}}\|_1) + \frac{\eta S^2}{2} + \rho\right)}\left(1 + \sqrt{\frac{\beta_g}{\tilde{\mu}_g}}\right) \tag{67}$$

which concludes the proof.

### A.16    ADDITIONAL EXPERIMENTAL DETAILS

We adopt the same data pre-processing steps as in (Chamon & Ribeiro, 2020) and use a two-layer neural network with 64 nodes and sigmoid activations. The counterfactual fairness constraint upper bound is set to $0.001$. We train this model over $T = 400$ iterations using a ADAM, with a batch size of 256, a primal learning rate equal to $0.1$ and weight decay magnitude set to $10^{-4}$. The dual variable learning rate is set to 2.

