# OpenReview forum: "Near-Optimal Solutions of Constrained Learning Problems"
_ICLR.cc/2024/Conference — ICLR 2024 poster_

### Official Review · Reviewer_z2M9 · 2023-10-20

**Soundness:** 2 fair
**Presentation:** 3 good
**Contribution:** 2 fair
**Rating:** 6
**Confidence:** 3

**Summary:**

This paper studies a parameterized constrained learning problem and proves that the infeasibility of the Lagrangian minimizer as well as that of the best iterate of dual ascent algorithms is close to that of the corresponding unparameterized convex constrained learning problem. Such a gap is proportional to the parameterization error, which means the Lagrangian minimizer of the parameterized constrained learning problem is nearly feasible under rich parameterization. Compared with the randomized or averaged sequence of primal iterates studied in the existing theoretical results, the best primal iterate studied in this theoretical work is more practical.

**Strengths:**

Originality: This work provides original theoretical results as explained in my summary above. In addition, the proof techniques in Section 3.3 look novel.

Quality: The theoretical error bound proportional to the parameterization error is reasonable by intuition.

Clarity: The general structure of this paper is clear.

Significance: The best iterate is more practical than the commonly used randomized or averaged sequence of primal iterates, so it is meaningful to provide the first convergence analysis of the best iterate.

**Weaknesses:**

(1) The theory could be improved in the following aspects:

(1.1) In line 4 of Algorithm 1, the **exact** minimizer of this unconstrained **nonconvex** optimization problem cannot be obtained. This minimization error should be considered which I envision will involve the parameterization error $\nu$ plus the error rate of the corresponding convex unconstrained optimization. In addition, for the stochastic version of Algorithm 1, the error of $\widetilde{\ell}_i\approx \ell_i$ should also be considered in line 4, not just in the dual update step in line 5.

(1.2) The error of $f_{\theta}(t)$ as well as $\widetilde{\ell} _ i\approx \ell _ i$ will also cause the approximation error of $g_p(\lambda(t))\approx\widetilde{\ell} _ 0[f _ {\theta}(t)]+\lambda(t)^{\top}\widetilde{\ell}[f _ {\theta}(t)]$ for selecting the best prima iterate. This should also be considered.

**The above two are the most important problems I think. I would like to increase my score once they are addressed.**

(1.3) You could also obtain the difference of $\ell_0$ between parameterized and convex problems in both Theorem 3.5 and Proposition 4.2 so that the solution of the unparameterized problem is not only nearly feasible but also nearly optimal.

(1.4) You could obtain finite time convergence result, i.e. when replacing $\lambda^{\rm best}$ in Proposition 4.2 with $\lambda _ T^{\rm best}$, the best iterate up to a fixed iteration $T$, what is the convergence rate involving $T$? I think this is possible by using the existing convergence rates on the convex constrained optimization problems in terms of the constraint violation (like $\sum _ {i=1}^m \alpha _ i\max(\ell _ i, 0)$ for some constants $\alpha _ i>0$) as well as optimality. Then we upper bound difference of the convergence rates between the convex and parameterized versions.

(2) A few points are to be clarified as listed in the questions below.

**Questions:**

(1) About the example of Counterfactual Fairness Constraints

(1.1) Is this example proposed by you? If not, cite the sources of the model and data. Otherwise, say that this example is proposed by you.

(1.2) What do the constraints mean in this example? You could add an explanation to the paper.

(1.3) Should we use $\mathbb{E} _ {x,y}$ in both objective and constraints, with the same distribution of $(x,y)$?

(1.4) Why are some $x$ bolded and some are not? Are they the same $x$?

(1.5) $f _ {\theta}(x)$ and $f_{\theta}(x,z)$ have inputs of different dimensionality. How can you make that happen, for example, in a certain neural network with a certain fixed parameter $\theta$?

(2) At the end of Section 2, you mentioned that Lagrangian minimizers are not unique and some of them could contain infeasible optimal primal variables. Is this found by you or previous literature? If it is found you, you could provide examples to support your claim and emphasize this claim as one of your novelty. Otherwise, you could cite the papers that prove this claim.

(3) Right after the problem $(P_u)$ at the beginning of Section 3.1, you could mention $\phi\in\mathcal{F}$.

(4) It seems that $\phi(\lambda_u)$ (in Assumption 3.2) and $\Phi^*(\lambda_u)$ are the same, right? If yes, you could unify the notations. If not, you could explain the meaning of $\phi(\lambda_p^*)$ in the paper.

(5) In paragraph 2 of Section 3.2, it may be clearer to change ''We will first provide a result assuming this curvature is known'' to something like ''We will first provide a result with the following assumption on the knowledge of this curvature'', so readers can relate to Assumption 3.4.

(6) After ''The second one captures the effect of parametrizing the hypothesis class for a fixed dual variable.'', you could add something like ''We will elaborate these perturbations in Section 3.3.1 and Section 3.3.2 respectively.''

(7) In the experiment,

(7.1) What negative likelihood function $\ell _ 0$ do you use (or the distribution of $x, y$)? What function $f _ {\theta}$ do you use?

(7.2) What is the math expression of the counterfactual fairness in the middle of Figure 2?

---

> ### Author Response · Authors · 2023-11-20
> **Response to reviewer z2M9**
>
> We thank the reviewer for this time and detailed feedback.
>
> **(1.1)** The effect of solving line 4 approximately is quite mild, as considered in Theorem 2 of [7]. Essentially, it introduces a bias on the supergradient used to update $\lambda$ in line 5. While this bias limits the precision at which the $\lambda_p^\star$ can be recovered, its effects do not accumulate over the trajectory. Indeed, suppose we replace line~4 with a $\rho-$approximate oracle, i.e., one capable of finding $f\_{\theta}^{\dagger} \in \mathcal{F}\_{\theta}$, such that
>
> $$
> L(f\_{\theta}^{\dagger}, \lambda_p) \leq \min\_{\theta\in\Theta} L(f\_{\theta}, \lambda_p) + \rho
> $$
>
> for $\rho \geq 0$. Then, $\ell(f_{\theta}^{\dagger})$ is an  approximate supergradient of $g_p$, i.e.,
>
> $$
> g_p(\lambda) \geq g_p(\lambda_p) + (\lambda - \lambda_p)^T \ell(f_{\theta}^{\dagger}) - \rho
> $$
>
> Hence, Lemma 4.1 holds with an extra term depending on $\rho$, namely,
> $$
> \lim_{t \to \infty} g_p^{\text{best}}(t | \lambda(t_0)) \geq D^{\star}_p - \left(\frac{\eta S^2}{2}+\rho\right) \quad \text{a.s.,}
> $$
>
> which propagates to Proposition 4.2. We will extend our results to account for a~$\rho-$approximate oracle in the revised manuscript.
>
> It is worth noting that, even in the non-convex case, there is a myriad of results showing that near-optimal solutions (small $\rho$) can be obtained for line 4 using gradient descent, e.g., for overparametrized neural networks [3,4,5,6]. Since this approximation affects the results in a controlled manner, we hope that this addresses the reviewer's concern.
>
> **(1.2)** This issue is related to the estimation error (i) that we mention in our conclusion as potential future work as well as in our overall response above when describing the context of our paper. As we explain there, the focus of this paper is on analyzing the algorithmic error (iii) that arises from retrieving a specific model from Alg. 1 rather randomizing over its iterates. We then argue that considering other sources of error (such as the estimation error) is beyond the scope of this paper considering that (a) the approximation and estimation errors were already analyzed in detail in [7] (Section 3) and (b) analysis of the algorithmic error already involves considerable challenges and novel proof techniques (as noted by the reviewer as well as Y13X and rUm3). Naturally, we welcome further discussions with the reviewer if they believe the paper contributions are not sufficient.
>
> That being said, we provide a potential roadmap (following [7], Prop 3.4) extending our results in order to show that estimation error, though non-trivial, is not impossible.
>
> [7] tackled this issue in the context of the duality gap between an empirical version of $(\text{D}_p)$ and $(\text{P}_p)$ using uniform convergence. Indeed, many results from learning theory (such as those based on VC dimension or Rademacher complexity) establish that, with probability $1-\delta$ over sample draws of $N$ samples,
>
> $$
> \left|\mathbb{E}\_{\mathfrak{D}}\left[\ell_i\left(f\_{\theta}(x), y\right)\right]-\frac{1}{N} \sum_{n=1}^N \ell_i\left(f\_{\theta}\left(x_n\right), y_n\right)\right| \leq \zeta_i(N, \delta),
> $$
>
> for all $\theta \in \Theta$ [18]. The function $\zeta_i(N, \delta)$ establishes the sample complexity of the problem. Under uniform convergence, every optimization problem can be solved within $\zeta_i(N, \delta)$ (with high probability). The implications of this were characterized in [7] Theorem 1, where they proved a near-feasible primal solution exists, **but not that it can be recovered** without randomization as we do here (see [7] Theorem 3).

---

> > ### Author Response · Authors · 2023-11-20
> > **Response to reviewer z2M9 (continued)**
> >
> > **(1.3)** As we mention in the general response, the near-feasibility results for $f\_{\theta}(\lambda^{\star}_p)$ derived in this work can be combined with the duality gap bound from [7], [15] to bound the near-optimality $|\ell_0(f\_{\theta}(\lambda^{\star}_p)) - P^{\star}|$. Namely, as long as $P^{\star}_p - D^{\star}_p \leq \Delta$, it holds that
> >
> > \begin{equation}
> > \tag{1}
> > | P_p^\star - \ell_0(f\_{\theta}(\lambda^{\star}_p))| \leq \\| \lambda^{\star}_p \\|_1 \\| \ell(\phi^{\star})  - \ell(f\_{\theta}(\lambda^{\star}_p)) \\| \_{\infty} + (1 + \|\lambda^{\star}_p \|_1 ) M\nu + \Delta
> > \end{equation}
> >
> > Taking $\Delta$ from [7] Proposition 3.3, and the bound in our Corollary 3.9 yields the desired result. Note that, plugging these values, (1) is of the same order as Corollary 3.9, i.e., no new terms appear, only the constants are different. We will include this bound as a corollary in the revised version of the manuscript. For the reviewer's reference, we provide a sketch of the derivations below.
> >
> > Using the fact that $P^{\star}_p - D^{\star}_p \leq \Delta$ and $P^{\star}_u \leq P^{\star}_p$, we obtain that
> >
> > $$
> > P_u^\star - D_p^\star \leq \Delta \Leftrightarrow
> > \ell_0(\phi^\star) - \ell_0(f\_{\theta}(\lambda^\star_p)) \leq \lambda^{{^\star}T}_p \ell(f\_{\theta}(\lambda^\star_p)) + \Delta
> > $$
> >
> > Since $\phi^*$ is feasible [$\ell(\phi^\star)) \leq 0$] and $\lambda^\star_p \succeq 0$ we obtain
> >
> > $$
> > \begin{aligned}
> > \ell_0(\phi^\star) - \ell_0(f\_{\theta}(\lambda^\star_p)) & \leq \lambda^{{\star}T}_p  (\ell(f\_{\theta}(\lambda^{\star}_p) -\ell(\phi^{\star})) + \Delta  \\\
> > & \leq \Vert \lambda^\star_p \Vert_1  \Vert \ell(\phi^{\star}) - \ell(f\_{\theta}(\lambda^{\star}_p)) \Vert\_{\infty} +  \Delta
> > \end{aligned}
> > $$
> >
> > To conclude the derivation of (1), we use $\nu$-universality and $M$-Lipschitz continuity (Assumptions 3.1 and 3.3) to get that there exists $\theta^\prime$ such that $\vert \ell_i(\phi^{\star}) - \ell_i(f\_{\theta^\prime})\vert \leq M\nu$. The duality gap bound then implies the approximate saddle-point relation
> >
> > $$
> > L(f\_{\theta}(\lambda^\star_p), \lambda^\star_u) - \Delta \leq L(f\_{\theta}(\lambda^\star_p), \lambda^\star_p) \leq L(f_{\theta^\prime}, \lambda^\star_p) + \Delta,
> > $$
> >
> > whose second inequality gives
> >
> > $$
> > \begin{aligned}
> > \ell_0(\phi^\star) - \ell_0(f\_{\theta}(\lambda^\star_p)) &\geq \ell_0(f\_{\theta^\prime}) - M\nu - \ell_0(f\_{\theta}(\lambda^\star_p))
> > \newline
> > & \geq \lambda^{{\star}T}_p (\ell(f\_{\theta}(\lambda^\star_p)) - \ell(\phi^\star)) - \Delta - (1 + \Vert \lambda^{\star}_p\Vert_1 )M\nu
> > \newline
> > & \geq - \Vert \lambda^{\star}_p \Vert_1 \Vert \ell(\phi^{\star}) - \ell(f\_{\theta}(\lambda^{\star}_p)) \Vert\_{\infty} - (1 + \Vert \lambda^{\star}_p\Vert_1 ) M\nu - \Delta
> > \end{aligned}
> > $$
> >
> > which completes the proof.
> >
> > **(1.4)** Indeed, finite time convergence can be obtained because the maximization of the dual function is a concave maximization problem. Note that the dual function $g_p(\lambda)$ is concave, irrespective of whether problem $(P_p)$ is convex, so the procedure described in Algorithm 1 to maximize $g_p$ is a standard projected supergradient ascent algorithm, well-studied in the literature [12], [19].
> >
> > Projected dual supergradient ascent algorithms have a quadratic rate of convergence. That is, when using a diminishing step size, for $\lambda_T$ to be $\delta$-optimal,  $O(1/\delta^2)$ iterations are needed ([12] Section 2.4, [20]).
> >
> > Despite the fact that when using a diminishing step-size, $\lambda^{\text{best}}_t$ gets arbitrarily close to $\lambda^{\star}_p$, and, $f\_{\theta}(\lambda(t))$ gets arbitrarily close to **the set** of Lagrangian minimizers $\mathcal{F}(\lambda^{\star}_p)$, the difficulty lies in establishing which element of that set the iterates are converging towards. This is what we establish in this work. Although Section 4 already translates Theorem 3.5 into an algorithimc result, we have not fully explored the dynamics of Algorithm 1, which is a promising topic of future work.

---

> ### Author Response · Authors · 2023-11-20
> **Response to questions 1-2 of reviewer z2M9**
>
> **(1.1)** Counterfactual fairness was not defined by us. The definition was proposed and studied in [17] (see Definition 5). We use a constrained learning formulation to tackle that problem using the COMPAS dataset [21] as in [7], [15], [22].
>
> **(1.2)** In this setting, each constraint represents a requirement that the output of the classifier be near-invariant to changes in the protected features (in our case gender and race). For instance, the output of the classifier should be (almost) the same whether, for all other variables kept the same, the gender of a sample is changed from male to female (and vice-versa) or whether the race is changed from caucasian to african-american, latino to caucasian...
>
> **(1.3)** While our formulation does not require objective and constraints to be defined over the same distribution, this is how the requirement of counterfactual fairness is formulated (see Definition 5 in [17]).
>
> **(1.4-1.5)** The lack of boldface in the $x$ of the constraints of the learning problem are a typo. Effectively, all $x$ in the problem refer to the feature vector $\mathbf{x}$. As for the input of the function, in retrospect we must agree with the reviewer that our abuse of notation causes confusion. In an attempt to highlight the protected features, we collect them into the separate vector $\mathbf{z}$. In other words, $\mathbf{x} = [\bar{\mathbf{x}}; \mathbf{z}]$, so that we can write $f\_{\theta}(\mathbf{x}) = f\_{\theta}(\bar{\mathbf{x}}, \mathbf{z})$. This is more in keep with the notation used in fair learning rather than the invariant learning. But this distinction is unnecessary as we could define counterfactual fairness more generally, with respect to any transformation of the data, i.e., for a set of transformations~$\{\rho_i\}$,
> $$
> \mathbb{E}\_{(x, y)} \left[ D_{KL}(f\_{\theta}(\mathbf{x}) || f\_{\theta}(\rho_i (\mathbf{x})) \right] \leq c, \: \text{for all } i.
> $$
> If the reviewer agrees with us that this is less prone to confusion, we will reformulate the problem in those terms in the revised manuscript.
>
> **(2)** We would certainly not claim this as contribution of the manuscript as it is a classical result from duality theory [23], [24], [16]. A straightforward example where this happens is in linear programs, where the Lagrangian evaluated at the optimal dual variable $L(\phi, \lambda^\star)$ is a constant (independent of $\phi$). In this case, $\text{argmin}\_{\phi \in \mathcal{F}}\ L(\phi, \lambda^\star) = \mathcal{F}$, which includes many infeasible candidates. This is why, as we mentioned our general response, it is not always possible to extract a \emph{feasible solution} for $(P_p)$ from the optimal dual variables $\lambda^\star_p$ without using primal averaging (for convex problems only) or randomization (for both convex and non-convex problems). Figure 1 (left) in our work further illustrates this phenomenon: though the blue constraint is often infeasible, it turns out to be feasible on average.

---

> > ### Author Response · Authors · 2023-11-20
> > **Response to questions 3-7 of reviewer z2M9**
> >
> > **(3)** We will highlight this fact in the revised manuscript.
> >
> > **(4)** Rather than constantly writing "for $\phi \in \Phi^\star(\lambda)$," we use the shorthand~$\phi(\lambda)$ to denote any element of the set of Lagrange minimizers $\Phi^\star(\lambda)$. In the unparametrized problem, the Lagrangian minimizer is unique, so $\phi^{\star} = \phi^\star(\lambda_u) = \Phi^\star(\lambda_u)$. We agree that this notation can be confusing and that it needs to be further emphasized and explained in the manuscript. We will carefully review the notation in the revised version of the paper.
> >
> > **(5-6)** We appreciate the reviewer suggestions and will incorporate these together with additional clarifications in our revision.
> >
> >
> > **(7.1)** We use the multinomial log-likelihood (as shown when we introduced the counterfactual fairness problem), also known as the cross-entropy loss. We will fix this typo and clarify the loss function in the revised manuscript. As we state at the beginning of Section 5, we take $f\_{\theta}$ to be a "two-layer neural network with 64 nodes and sigmoid activations."
> >
> > **(7.2)** In Figure 2(middle), we show the proportion of test samples whose predictions change when the indicated modification is made to the protected variables. Explicitly,
> > $$
> > \text{cf}\_i(f\_{\theta}) = \frac{1}{N} \sum_{j=1}^N \mathbf{1}\left[ \text{argmax}(f\_{\theta}(\mathbf{x_j}, \mathbf{z_j})) = \text{argmax}(f\_{\theta}(\mathbf{x_j}, \rho_i(\mathbf{z_j}))) \right]
> > $$
> > It is to counterfactual fairness what accuracy is to classification. We will modify the axis label and specify this more clearly in the text.

---

> > > ### Comment · Reviewer_z2M9 · 2023-11-21
> > > **Raise my rating to 6.**
> > >
> > > I have raised my rating to 6 as the authors make it clear what approximation error to focus on, and give explanation to those uncovered approximation errors.
> > >
> > > Just one suggestion:
> > > You could incorporate the citations in your answer to my questions (1-2) into your paper.
> > >
> > > Thanks for your elaborate explanation.
> > >
> > > Reviewer z2M9

---

### Official Review · Reviewer_rUm3 · 2023-10-22

**Soundness:** 3 good
**Presentation:** 4 excellent
**Contribution:** 3 good
**Rating:** 8
**Confidence:** 3

**Summary:**

This paper discusses the primal dual algorithm for training a neural network. Since the constraint is difficult to handle, one typical alternative solution is to use Lagrange multiplier to penalize the constraint violation in the objective function and calculates its minimum. This paper specifically calculates and bounds the violation of the constraint to demonstrate that the primal dual algorithm is still good for training NN. Experiments also verify the theoretical conclusion.

Update:
I read the rebuttal. I found the contribution that this paper bounds the optimality gap which is not generally known in convex case, and the way it handles constraint and shows that the primal convergence, which does not in nonconvex case in general, is important. I raise the score to 8 while not having checked all details.

But I still wonder how good the approximation of the functional class is, i.e., how big is the distance from any function to a function in the parameterized family that the training algorithm is in, and how training error indicates the test error when one has finite samples but a parameterized family to train on being very large. With that, I do not raise confidence and do not raise score to 10.

**Strengths:**

I think this paper is mathematically correct, detailed, and self consistent. The writing is clear, which gives a lot of discussion, and the logic and the structure that goes to the final conclusion step by step is reader friendly. The analysis of different scenarios under the big primordial framework is comprehensive and detailed.

**Weaknesses:**

Besides finding the extent of violation of the constraints, it is also very important to bound the objective function’s suboptimality of the final iterates. Especially, since this paper is positioned in NN scenario, the objective function is typically highly nonconvex, and there might be some optimality gap when introducing the dual function, which is relevant to the convex hull of the primal function rather than the primal function itself.

**Questions:**

This paper introduces an algorithm which has two inner oracles the first optimizes the Lagrangian function by an argmin expression, while the dual variable part is easy to understand. Is this a typical method when we train your networks under constraints, or one typically use the gradient projection? I guess there is a large difference between the sub optimality gap of these two algorithms, so that the authors prefer Algorithm 1. What is the calculation of complexity of the oracle Line 4, and what is the total convergence rate and complexity of algorithm 1?

How to solve Line 4? If you solve it multiple times, what is the total complexity?

How to choose $\eta$?

Page 3, "... $P_p$ is convex" maybe use "... eq$(P_p)$ is convex" to differentiate $P_p$ and $P_p^*$.

I think people in ML area are not familiar with functional optimization so it would be great to explain more. For example, Assumption 3.1, it can be "$M$-Lipschitz \emph{with respect to functional $\ell_2$ norm}" with a footnote about the norm. Also explain strong convexity and smoothness in functional space.

Is there an interpretation of Assumption 3, or for common parameterized classes of functions, can you give an example how large the gap is?

Thm. 3.5, I think the $\ell_2$ norm on left hand side is a two norm on scalar right? Can you just use $(l(...) - l(...))^2$ or $|l(...) - l(...)|$? Typically we only write norm for vectors.

---

> ### Author Response · Authors · 2023-11-20
> **Response to reviewer rUm3**
>
> We thank the reviewer for acknowledging the soundness, quality of presentation and significance of the contributions of this work. In what follows, we address the weaknesses and answer the concerns raised by the reviewer.
>
> **(1)** The reviewer is correct in that, unlike the unparametrized problem $(P_u)$ which is strongly dual, the constrained learning problem $(P_p)$ will display a positive duality gap ($P_p^\star - D_p^\star$). However, [7] Proposition 3.3 shows that this gap is bounded under our Assumptions 3.1-3.3 (see details general response). As also pointed by reviewer z2M9, this duality gap bound can be combined with the results from our paper to bound the near-optimality $|\ell_0(f_{\theta}(\lambda^*_p)) - P_p^\star|$. Namely, for $P^*_p - D^*_p \leq \Delta$, it holds that
>
> \begin{equation}
> \tag{1}
> | P_p^\star - \ell_0(f\_{\theta}(\lambda^{\star}_p))| \leq \\| \lambda^{\star}_p \\|_1 \\| \ell(\phi^{\star})  - \ell(f\_{\theta}(\lambda^{\star}_p)) \\| \_{\infty} + (1 + \|\lambda^{\star}_p \|_1 ) M\nu + \Delta
> \end{equation}
>
> Taking $\Delta$ from [7] Proposition 3.3, and the bound in our Corollary 3.9 yields the desired result. Note that, plugging these values, (1) is of the same order as Corollary 3.9, i.e., no new terms appear, only the constants are different. We will include this bound as a corollary in the revised version of the manuscript. For the reviewer's reference, we provide a sketch of the derivations below.
>
> Using the fact that $P^{\star}_p - D^{\star}_p \leq \Delta$ and $P^{\star}_u \leq P^{\star}_p$, we obtain that
>
> $$
> P_u^\star - D_p^\star \leq \Delta \Leftrightarrow
> \ell_0(\phi^\star) - \ell_0(f\_{\theta}(\lambda^\star_p)) \leq \lambda^{{^\star}T}_p \ell(f\_{\theta}(\lambda^\star_p)) + \Delta
> $$
>
> Since $\phi^*$ is feasible [$\ell(\phi^\star)) \leq 0$] and $\lambda^\star_p \succeq 0$ we obtain
>
> $$
> \begin{aligned}
> \ell_0(\phi^\star) - \ell_0(f\_{\theta}(\lambda^\star_p)) & \leq \lambda^{{\star}T}_p  (\ell(f\_{\theta}(\lambda^{\star}_p) -\ell(\phi^{\star})) + \Delta  \\\
> & \leq \Vert \lambda^\star_p \Vert_1  \Vert \ell(\phi^{\star}) - \ell(f\_{\theta}(\lambda^{\star}_p)) \Vert\_{\infty} +  \Delta
> \end{aligned}
> $$
>
> To conclude the derivation of (1), we use $\nu$-universality and $M$-Lipschitz continuity (Assumptions 3.1 and 3.3) to get that there exists $\theta^\prime$ such that $\vert \ell_i(\phi^{\star}) - \ell_i(f\_{\theta^\prime})\vert \leq M\nu$. The duality gap bound then implies the approximate saddle-point relation
>
> $$
> L(f\_{\theta}(\lambda^\star_p), \lambda^\star_u) - \Delta \leq L(f\_{\theta}(\lambda^\star_p), \lambda^\star_p) \leq L(f_{\theta^\prime}, \lambda^\star_p) + \Delta,
> $$
>
> whose second inequality gives
> $$
> \begin{aligned}
> \ell_0(\phi^\star) - \ell_0(f\_{\theta}(\lambda^\star_p)) &\geq \ell_0(f\_{\theta^\prime}) - M\nu - \ell_0(f\_{\theta}(\lambda^\star_p))
> \newline
> & \geq \lambda^{{\star}T}_p (\ell(f\_{\theta}(\lambda^\star_p)) - \ell(\phi^\star)) - \Delta - (1 + \Vert \lambda^{\star}_p\Vert_1 )M\nu
> \newline
> & \geq - \Vert \lambda^{\star}_p \Vert_1 \Vert \ell(\phi^{\star}) - \ell(f\_{\theta}(\lambda^{\star}_p)) \Vert\_{\infty} - (1 + \Vert \lambda^{\star}_p\Vert_1 ) M\nu - \Delta
> \end{aligned}
> $$
> which completes the proof.

---

> > ### Author Response · Authors · 2023-11-20
> > **Response to questions 1-3 of reviewer rUm3**
> >
> > **(1)** It is not clear what the reviewer means by "algorithm which has two inner oracles." Alg. 1 depends on a single oracle (namely, Line 4), which corresponds to a regularized, unconstrained learning problem. Such an oracle is typical in this type of analysis (see, e.g., [7], [10], [11]). Note that the guarantees in all these works require not only on the existence of an oracle (as in Line 4), but also randomizing over the algorithm iterates (see [10], Theorem 2; [11], Theorem 4.1; [7], Theorem 3]). The main contribution of this paper is to prove conditions under which it is possible to do away with such impractical randomization procedures in non-convex problems.
> >
> > As the reviewer notes, the non-convexity of the constraints hinders the use of many constrained optimization methods, such as gradient projections. Indeed, there is no straightforward way to compute such projections, so that projecting over the feasibility set is essentially as hard as solving the original constrained learning problem. While this also affects the complexity of the oracle in Line 4, it remains an unconstrained optimization problem (as opposed to a constrained one, such as those involved in computing projections). Hence, it is akin to training a (regular) ML model, for which SGD can often find good minimizers even in the non-convex setting (e.g., for overparametrized NN [3,4,5,6]). Note that, as we explain in our response to reviewer z2M9, relaxing Line 4 to an *approximate* oracle only has a mild effect on the bound, which we will account for in the revised version of the manuscript.
> >
> > Finally, the main goal of our paper is to study the convergence of primal iterates, which as we outline in the general response, is not a given. Even in the case of convex optimization problems, it is not always true that primal iterates of Alg. 1 ($\theta_t$) converge to a optimal, feasible solution of the primal problem $(P_p)$. Our paper provides a novel analysis for (non-convex) learning problems which requires the development of novel analyses and proof techniques. Hence, while we agree that convergence rate results are interesting, we think they are the topic of future research.
> >
> > **(2)** The convergence guarantee in Lemma 4.1 explicitly shows the effect of $\eta$ on Alg. 1, i.e., Lemma 4.1 provides convergence guarantees of dual gradient ascent when applied to a problem with non-convex constraints. The result is the same as the convergence properties of dual gradient ascent for an optimization problem with convex constraints and is provided here for completeness. The reason why both results are identical is that the dual function $g_p(\lambda)$ is always concave, irrespective of the convexity of the constraints. Hence, the choice of $\eta$ does not differ significantly from other gradient methods, i.e., one may use a diminishing step size to converge to the optimum (as long as $\sum_{t=1}^{\infty} \eta_t^2 < \infty$ and $\sum_{t=1}^{\infty} \eta_t = \infty$ [12]) or adaptive methods such as Adam (as was done in Section 4).
> >
> > Notice, however, that this only guarantees convergence in of the dual variable $\lambda$. For convex optimization problems, this implies that the primal iterates (strongly convex) or the average of the primal iterates (convex, but not strongly convex) converge to a neighborhood of the optimum. This is not the case for general non-convex problems, for which it is easy to build examples where primal iterates are arbitrarily far from optimal solutions due to the presence of a positive duality gap. In this paper, we show conditions under which such convergence can be guaranteed.
> >
> > **(3)** The manuscript does not assume knowledge of functional analysis and is self-contained, providing all definitions and properties required by our analyses. In particular, Lipschiptz continuity, smoothness, and strong convexity are defined as in finite dimensional optimization when using the definition of Fréchet derivative in Appendix A.1. Still, we agree that readability could be improved by grouping all required prior results. In the revised manuscript, we will aggregate all functional analysis results (as well as precise definitions of Lipschiptz continuity, smoothness, and strong convexity) in Appendix A.1.

---

> > > ### Author Response · Authors · 2023-11-20
> > > **Response to questions 4-5 of reviewer rUm3**
> > >
> > > **(4)** Assumption 3 is a measure of how good the parametrization or model is at representing functions. In other words, it determines if $\mathcal{F}\_\theta$ is a good cover of $\mathcal{F}$. Consider, for instance, an RKHS. Then, if $\mathcal{F}_\theta$ contains linear combinations of 10 kernels, it is considerably worse at representing functions in the RKHS than combinations of 1000 kernels. The same holds for the set of continuous functions and two-layer NNs with sigmoidal activation [13].
> > >
> > > The only difference between the unparametrized convex problem, which has a unique (feasible) solution $\phi^*$ and the non-convex constrained learning problem is the set over which the optimization is carried out. If the set is a good cover of $\mathcal{F}$, then we can expect the properties of the solutions to the underlying problems to be similar. We quantify this similarity in Assumption 3.3, through the constant $\nu$.
> > >
> > >  While determining the exact value of $\nu$ is in general not straightforward, we know for both of these examples that any $\nu > 0$ can be achieved for a large enough number of kernels or neurons.  In fact, existing results in both cases hold for sup-norm, which is much stronger than the $L_2$ approximation Assumption 3 requires. What our results show is that the quality of the output of Alg. 1 has mild dependence on $\nu$, so that better guarantees can always be obtained by increasing the size of these models.
> > >
> > > **(5)** Recall from Section 2.1 that $\ell$ is a vector-valued function, namely, $\mathbf{\ell}(\phi) = \left[ \mathbf{\ell}_1(\phi), \dots, \mathbf{\ell}_m(\phi) \right]$ for $\phi \in \mathcal{F}$. Hence, the norms in Theorems 3.5, Corollary 3.9, and Proposition 4.2 are vector norms.

---

### Official Review · Reviewer_2G7p · 2023-10-31

**Soundness:** 4 excellent
**Presentation:** 3 good
**Contribution:** 2 fair
**Rating:** 6
**Confidence:** 3

**Summary:**

This paper considered the problem of providing a theoretical guarantee on the feasibility of the last-iteration solution of a Dual Constrained Learning Algorithm (i.e., dual ascent algorithm applied to the constrained learning model parameterized by $\theta$). Typically, to get a feasible output of the dual acsent algorithm applied to the constrained learning models, one need to perform averaging or randomization over the whole output sequence, which is impractical in reality. Let $P_p$ denote the parameterized constrained learning problem where the candidate functional solutions satisfy $f_\theta\in F_\Theta$, $\theta\in\Theta$. In addition, let $P_u$ be the unparameterized constrained learning problem where the candidate functional solutions come from the set $F\supseteq F_\theta$. The main contribution of this paper is that authors showed that the optimal lagrangian minimizers of problem $P_p$, denoted by $f_\theta(\lambda^*_p)$, are close to the optimal lagrangian minimzer of problem $P_u$, $\phi(\lambda_u^*)$.

**Strengths:**

1. Let $\ell(f_\theta)\leq 0$ be the constraints that are required for the output $f_\theta$ to satisfy. The author showed for the first time that under some assumptions on the constraints $\ell$, it holds $||\ell(f_\theta(\lambda_p^*))- \ell(\phi)||$ is bounded by some constants, where $f_\theta(\lambda^*_p)$ is the optimal lagrangian min of $P_p$ and $\phi^*$ is the optimal lagrangian min of $P_u$ such that $\ell(\phi^*)\leq 0$.
2. The authors provided high level explanation and intuition which helps the reader to understand the paper better.

**Weaknesses:**

1. the bound on $||\ell(f_\theta(\lambda_p^*))- \ell(\phi)||$ contains $||\lambda_p^*||1$, which could be large when the number of coinstraints $m$ is large. When $||\lambda_p^*||$ is large, we cannot assert that $f_\theta(\lambda^*_p)$ is almost feasible. Is there any way to show that $||\lambda_p^*||_1$ is small under some assumptions?
2. The authors considered $f_\theta(\lambda_p^*)$, the optimal lagrangian minimzer of $P_p$. However, one can only get $f_\theta(\lambda_p^*)$ after infinite iterations. It would be more interesting if the authors can provide some information on the feasibility of $f_\theta(\lambda(T))$. In particular, how close can $f_\theta(\lambda(T))$ and $f_\theta(\lambda_p^*)$ be?
3. The author measured the distance between $\ell(f_\theta(\lambda_p^*))$ and $\ell(\phi^*)$. However, $\phi^*$ doesnot necessarily belong to $F_\theta$. It seems a feasible solution from $F_\theta$ should be a better option and perhaps will bring us better bounds.

**Questions:**

1. Can the authors provide some information about the convergence property of the dual-ascent algorithm applied to a problem with non-convex constraints?

---

> ### Author Response · Authors · 2023-11-20
> **Response to reviewer 2G7p**
>
> Thank you for your time and effort in reviewing our paper and for your positive evaluation of the soundness and presentation of our work. In what follows, we address the three weaknesses that were brought up.
>
> **(1)** Yes. Consider that there exists a parameter $\theta_0$ and a strictly positive constant $C>0$ such that for all constraints $i$ we have that $\ell_i(f_{\theta_0}) \leq -C$. We then have that
> \begin{equation}
> \tag{1}
> \\| \lambda^{*}_p \\|_1 \leq \frac{\ell_0(f\_{\theta_0})-D^{\star}_p }{C} .
> \end{equation}
> This is a standard result in duality theory [8] that we have verified holds for the nonconvex optimization problems that we study in this paper. We will incorporate this result and its proof in the final manuscript.
>
> We point out that the existence of $\theta_0$ and $C>0$ satisfying $\ell_i(f_{\theta_0}) \leq -C$ is not a new assumption. We already require strict feasibility in Assumption 3.2. We simply did not name the strict feasibility constant $C$. Further notice that the bound in (1) does not depend on the number of constraints involved in the optimization problem, but only on the properties of a strictly feasible point $\theta_0$.
>
> **(2)** This is a good point. Let us point out that the algorithm we propose in this paper is a standard stochastic dual gradient ascent algorithm in which dual iterates are moving along stochastic gradients of the dual function. Since the dual function is concave (dual functions are always concave for constrained minmization problems, convex or not), convergence properties of dual iterates are therefore well understood. They follow from standard convergence results of stochastic gradient ascent for concave functions [12], [20].
>
> In this dual gradient ascent algorithm primal iterates are a byproduct. They are computed as Lagrangian minimizers associated with dual iterates. The main goal of our paper is to study the convergence of these primal iterates, which, as we outline in the general response, is not a given. Even in the case of convex optimization problems, it is not always true that primal iterates of Alg. 1 converge to optimal primal variables.
>
> Our paper provides a novel analysis for the convergence of primal iterates in (nonconvex) learning problems. To do so we develop novel analyses and proof techniques relying on the comparison of the problem of interest with a convex reference given by the unparameterized constrained learning problem. We think that it is best for readers to keep the paper focused on this contribution and that, consequently, finite time results are a topic of future research.
>
> **(3)** This is an interesting point. The bounds that we provide in Theorem 3.5 and Corollary 3.9 are of the form:
>
> \begin{equation}
> \tag{2}
> |  \ell_i(f_{\theta}(\lambda^{\star}_p)) - \ell_i(\phi^{\star}) | \leq \zeta,
> \end{equation}
>
> for some specific constant $\zeta$. This equation is a comparison between the loss $\ell_i(f_{\theta}(\lambda^{\star}_p))$ obtained from the Lagrangian minimizers of the *parameterized* Lagrangian with the loss $\ell_i(\phi^{\star})$ of the solution of the *unparameterized* constrained learning problem.
>
> We take it that the reviewer expected to see a comparison between the loss $\ell_i(f\_{\theta}(\lambda^{\star}_p))$ obtained from the Lagrangian minimizers of the *parameterized* Lagrangian with the loss $\ell_i(f\_{\theta}^*))$ of the solution of the *parameterized* constrained learning problem. They are therefore intrigued by the bounds that appear in Theorem 3.5 and Corollary 3.9.
>
> The reason for the unexpected appearance of the difference in the left hand side of Corollary 3.9 is the core technical innovation of our proof techniques. The parametrized learning problem with variables $f_\theta$ is non-convex and difficult to analyze. The unparametrized problem with variable $\phi$ is convex and one that we can analyze using functional convex analysis techniques. Our technical approach relies on the fact that these two problems and their duals can be shown to be $O(\nu)$-close to each other if the parametrization is $\nu$-universal. This is the reason why feasibility bounds end up comparing the loss $\ell_i(f\_{\theta}(\lambda^{\star}_p))$ obtained from the Lagrangian minimizers of the *parameterized* Lagrangian with the loss $\ell_i(\phi^{\star})$  of the solution of the *parameterized* constrained learning problem -- as opposed to the loss $\ell_i(f\_{\theta}^{\star}))$ of the solution of the *parameterized* constrained learning problem.
>
> Further notice that since $\phi^*$ is feasible (i.e: $\ell_i(\phi^*)\leq0$) , an upper bound on the infeasibility of $f_{\theta}(\lambda^*_p)$ can be derived directly from (2). Explicitly, $\ell_i(f\_{\theta}(\lambda^*_p))  \leq  \ell_i(f\_{\theta}(\lambda^*_p)) - \ell_i(\phi^*)  \leq \zeta$.

---

> > ### Author Response · Authors · 2023-11-20
> > **Response to question of reviewer 2G7p**
> >
> > **(1)**
> >
> > This is a good point. Lemma 4.1 in our paper provides a convergence guarantee of dual gradient ascent when applied to a problem with non-convex constraints. The result is the same as the convergence properties of dual gradient ascent for an optimization problem with convex constraints and is provided here for completeness. The reason why both results are identical is that the dual function $g_p(\lambda)$ is always concave, irrespective of the convexity of the constraints.
> >
> > Notice that the same can't be said of the convergence of primal iterates. For convex optimization problems one can show that primal iterates converge to a neighborhood of the optimal variables when the optimization problem is strongly convex. When the problem is convex but not strongly convex, one can show convergence of the average of primal iterates. For general non-convex problems it is impossible to show convergence of primal iterates and, in fact, it is easy to build examples of problems where primal iterates are arbitrarily far from optimal solutions. This happens because of the presence of a positive duality gap.
> >
> > Our main contribution is to provide a convergence analysis of primal iterates for this non-convex optimization problem in which the richness of the parametrization implies its proximity to a convex optimization problem---the unparameterized constrained learning problem.

---

### Official Review · Reviewer_3wd8 · 2023-11-01

**Soundness:** 3 good
**Presentation:** 3 good
**Contribution:** 2 fair
**Rating:** 3
**Confidence:** 2

**Summary:**

The paper studies the problem of contrained learning problems using the dual learning algorithm. The authors provide feasiblity gap (for both the primal and dual variables) between the parameterized/unparameterized problems under certain regularity conditions. The authors then provide optimality gap for the dual learning algorithm, and numerical validations for the hypothesis that the feature space size impacts optimality gap according to the theory.

**Strengths:**

The paper is clearly written and easy to read. The authors make the assumptions clear and define the problem well. The theory for the parts I checked is sound.

**Weaknesses:**

1. The paper's title is near-optimal solutions which is misleading. The main results compare the stationary points of constrained/unconstrained problems as well as using the dual learning algorithm. I'd expect optimality in statistical sense where a lower bound is provided for rates of convergence/estimation error, instead of studying the limiting points error, which is less interesting to me.
2. Following the previous point, bounding the error of stationary points is often less interesting in terms of optimality, and lacks of theoretical contributions to the community. Perhaps the authors can provide why such analyses is nontrivial/fundamentally harder than other problems and why it is important to understand this first.
3. The numerical experiments seem weak. It is not suprising that there will be a gap and as the number of features grow the model approximation error is smaller. I'm not seeing how this directly corrobates the theory. Ideally I would like to see (i) a hypothesis emerged from the theory (ii) an experiment that validates the hypothesis.

**Questions:**

See weakness.

---

> ### Author Response · Authors · 2023-11-20
> **Response to reviewer 3wd8**
>
> We appreciate the feedback of the reviewer. We refer to our general response to all reviewers for clarifications of the main motivations, contributions, and challenges tackled by our work. We hope that our answers below, together with our general response, clarifies these points, but welcome any further questions they may have.
>
>
> **(1)** While we appreciate the reviewer's point, we respectfully disagree. While it is true that "optimality" has different meanings in statistics and optimization/algorithms, we do not see why the former should take any priority over the latter. Optimization plays as key a role in modern data science and machine learning as statistics, as evidenced by the overwhelming number of theoretical and empirical results that heavily rely on optimization arguments or improvements on optimization algorithms. We do not therefore believe that what we take "near-optimality" to mean in this manuscript, which is in line with standard definitions both in optimization and computer science (e.g., [14]), is in any way misleading. That being said, we are happy to consider any alternative suggestion the reviewer might have.
>
> **(2)** As explained in the general comment, two challenges must be overcome in this non-convex setting. The first one is the lack of strong duality (as pointed out by reviewer rUm3). In other words, the value $D_p^\star$ of the dual problem $(D_p)$ need not be a good approximation of the value $P_p^\star$ of $(P_p)$. The magnitude of this gap was characterized in [7] Proposition 3.3. Naturally, the quantities that characterize the feasibility of primal iterates are present in the duality gap bound of [7]. For example, the duality gap $ P^{\star}_p - D^{\star}_p $ depends linearly on the richness of the parametrization $\nu$.
>
> The second main challenge is that dual ascent methods such as Algorithm 1 (also referred to as best response) need not converge, and even it did, recovering a feasible solution from optimal dual variables may not be possible. As explained in section 2.2 and shown experimentally on Figure 1, the non-uniqueness of Lagrangian minimizers can lead to large oscillations in the feasibility of primal iterates. In the linear programming example of the general comment (which is a convex problem) the primal iterates obtained as a byproduct of the ascent method can severely violate the constraints. Indeed, in this case, the Lagrangian $L(\phi, \lambda^\star)$ is a constant (independent of $\phi$), i.e., $\Phi^\star(\lambda^\star) = \mathcal{F}$. The difficulty of recovering (near-)feasible primal solutions explains the need of randomized solutions (see
> e.g, ([10], Theorem 2; [11], Theorem 4.1; [7], Theorem 3). However, randomized predictors are impractical and do not solve problem $(P_p)$. Thus, it is highly relevant to determine conditions under which a near-feasible primal iterate, **which is also near-optimal** can be recovered. Also note that the analysis carried out in this paper involved novel proof techniques (as per reviewer z2M9), such as the use of the curvature of the perturbation function (see section 3.3.2).
>
>
> **(3)** We respectfully disagree. The experiments in this paper clearly illustrate the fact that richer parametrizations lead to better properties of Alg. 1 in terms of the constraint violation of its iterates. This behavior, suggested by our theorems, is not straightforward. Indeed, this paper analyses the optimization error when solving constrained learning problems and not the estimation or approximation error~(see general response). Hence, while it is indeed expected that richer parametrizations will lead to lower approximation error, it is certainly not straightforward that richer models make optimization easier. While better parametrizations do provide more feasible models to choose from, they also contain more infeasible ones. Finding solutions requires handling a much larger class of hypothesis, so that we would actually expected the optimization to be harder. Our results, however, prove that richer parametrizations make it easier for the algorithm to find a feasible solution. In fact, not only a feasible solutions, but one that is also near optimal, which is not straightforward (as per our previous answer). This conclusion from Proposition 4.2 is illustrated in Figure 2(right). Note that this plot does not represent the approximation (test) error, but the constraint violation. We will work on improving the description and axis labels of these plots to avoid confusion in the future.

---

### Official Review · Reviewer_Y13X · 2023-11-03

**Soundness:** 3 good
**Presentation:** 3 good
**Contribution:** 3 good
**Rating:** 6
**Confidence:** 4

**Summary:**

This paper considers the problem of constrained learning using the dual-ascent relaxation framework. The paper attempts to bound the error in the constraints of the relaxed solution using the dual variables. The approach is based on an infinite-dimensional (not parameterized) formulation/relaxation of the dual problem in which the bounds are obtained. This provides a bound on the error in the infinite-dimensional primal problems and the results are used to bound the error in the actual solution to the primal using the actual finite-dimensional dual ascent.

**Strengths:**

The narrative of the paper is initially quite clear and the approach is relatively novel and intuitive. The problems seems interesting. There are clear theorem statements and clearly stated assumptions. The mathematics is mostly rigorous and quite general.

**Weaknesses:**

The interpretation of the results are not clearly stated. In an attempt to be perfectly general, it is hard to state exactly how the results apply to any particular learning problem. One would think that algorithms based on neural networks would perform quite differently than a kernel-based approach using a single poorly chosen kernel. And yet the results don't seem to account for such differences in any way. Additionally, this makes it hard to understand if the 7 (mostly regularity) assumptions are valid and what the corresponding parameters are.

**Questions:**

Specific concerns and questions are listed as follows.

1 Please more clearly spell out the motivation in the context of fairness of including the constraint vs. just not including data on the protected classes.

2 "$\mathcal{F}_\theta$ can be a neural network" -- First, its a set, not a network. But remind me -- in this case will the hypothesis space be convex? Also, take the extreme case of a single arbitrarily chosen Gaussian kernel. Is there not some problem with fragility/generalizability? Performance on untrained data? How does this approach account for the robustness of the parameterization? Maybe this relates to the question of which topology is being used to characterize Lipschitz continuity.

3 The Lipschitz and convexity are defined using $\ell$ which is a statistical function. Would it not be more suitable to define them using the actual functions, $\tilde{\ell}$?

4 Not much mention of the primal-dual gap and whether it is significant in parameterized or unparameterized problems

5 Generally bothersome to have lots of assumptions unless it can be shown they are satisfied in certain basis and useful cases.

6 I'm not sure exactly the point of Section 3.3.2 -- Are we saying that the bounds can be improved by making the constraints strict?

7 "We will denote by $\hat \ell$ an estimate of $\ell$ using the dataset D...." First, redefining $D$ casuses confusion. Also, how is this now a dataset? Also, this brings up the $\ell$ vs $\tilde{\ell}$ issue again -- Alg 1 is defined using $\ell$ and not $\tilde{\ell}$. What is $D^*_p$ in Lemma 4.1 -- clearly not related to D.... Actually it seems there are lots of $D$-based notation in this paper.

8  "We train this model over T = 400 iterations using a ADAM"  Acronym undefined -- maybe they mean Adam, which is an algorithm. What is $\ell$ here? Are the Assumptions satisfied? What are the obtained bounds?

Minor corrections are as follows

- Page 1 - "In fact, this problem is even hard from" -- which problem?

- Notation $D_{KL}$ on Page 2 is undefined.

- "guarantees usually pertain a probability distribution over"

- Assumption 3.2 is unclear. What is $f_\theta(\lambda)$ here? It becomes more difficult to interpret the results after this.

- "the unparametrized Lagrangian minimizer is unique" -- presumably this is the solution for the unparameterized version, $D_u$?

- "unparametrtized"

- Assumption 3.4 seems difficult to verify...

- Thm 3.5. This is difference between the constraint functions, not their violations since $\ell_i$ are not non-negative.

- Page 12, after Defn A.1. Some conflation of operator $B \in \mathcal{B}$ and its representor, $g$.

- Defn A.3 -- what does it mean for $h(x)>-\infty$?

- "Linear independence constraint qualification (LICQ)" -- capitalization issue...

- "Thus, it can be thought of as the baseline effect" --  define "it"... What is the third component -- I see only two.

- " Combining the bound in equation in equation 6 with"

- numerous problems with bibliography entries.

- Proofs in appendix need better organization. No explanation or connecting verbiage is given. For example, Appendix A.6

---

> ### Author Response · Authors · 2023-11-20
> **Response to reviewer Y13X**
>
> We thank the reviewer for the detailed feedback and for highlighting the novelty and rigour of our work. Next, we address the weaknesses they point out (i.e, interpretation of the results and strictness of assumptions) and answer the concerns they raised.
>
> **Weaknesses**
>
> **(1)** As we mention in our general response, the issue of neural networks vs poorly chosen kernel relates to the approximation (ii) [and perhaps generalization (iii)] error. In this manuscript, we study the algorithmic error (iii) that only considers how good a solution of $(P_p)$ can be obtained using Alg. 1. Whether $(P_p)$ yields good solutions for learning problem, i.e., errors (ii)-(iii), is tackled in, e.g., [7], [15]. In the case of a single poorly chosen kernel, there is no algorithmic error (iii): either the one kernel is feasible or the problem has no solution. This is reflected in our results ($\lambda \to \infty$ if $(P_p)$ is infeasible).
>
> Things are more interesting in the neural network case. Indeed, while we do not analyze the approximation error (ii), the richness of the parametrization still plays a role in our results. Indeed, we show that more expressive models (Assumption 3) have a positive impact on the algorithm behavior. In other words, there are stronger guarantees for Alg. 1 to find near-feasible solutions of $(P_p)$ when using larger NNs. This point is illustrated in Figure 2(right), which shows the constraint violation as a function of the model capacity. This is not straightforward: while better parametrizations do provide more feasible models to choose from, they also contain more infeasible ones. Finding feasible solutions requires handling a much larger class of hypothesis, so we could actually expected the optimization to be harder. Combined with the duality gap bounds from [7], [15], this implies that Alg. 1 finds solutions that are not only near-feasible, but also near-optimal. This is not straightforward considering the lack of strong duality due to non-convexity and the challenges in proving convergence of primal iterates without randomization (see general response).
>
> We will add new discussions of these interpretations and highlight these conclusions in the revised version of the manuscript.
>
> **(2)** Our results effectively depend on five assumptions (since Assumption 3.4 is implied by 3.1, 3.6, and 3.7 and number 3.5 is a theorem). We summarize them below:
>
> - **Assumption 3.1**: $\ell_i$ are convex and Lipschitz continuous. $\ell_0$ is strongly convex
> - **Assumption 3.2**: there exists a strictly feasible $\phi$, i.e., $\ell_i(\phi) \leq C$ for all~$i$ and some $C > 0$
> - **Assumption 3.3**: For all $\phi \in \mathcal{F}$, there exists $\theta \in \Theta$ such that $\| \phi - f_{\theta} \|_{L_2} \leq \nu$
> - **Assumption 3.6**: $\ell_i$ are smooth (Lipschitz gradients)
> - **Assumption 3.7**: The Jacobian $D_{\phi} \mathbf{\ell}( \phi^{\star})$ has full-row rank, where $D_{\phi}$ denotes the Fréchet derivative
>
> As the reviewer notes, these assumptions are mostly regularity assumptions over which a practitioner has full control. Indeed, Assumptions 3.1 and 3.6 impose mild restrictions on the losses, namely, that they be Lipschitz continuous, convex, and smooth. These are widely used to prove convergence results for gradient descent algorithms even in the convex setting and are satisfied by typical losses used in ML when, e.g., the hypothesis class $\mathcal{F}$ is compact. Assumption 3.3 is always satisfied for some finite $\nu$ and describes the richness of the parametrization. A detailed description of this assumption was provided in section 3.1 as well as our previous answer.
>
> Assumptions 3.2 and 3.7 are closely related to constraint qualifications widely used in the analysis of Alg. 1 in convex settings. Indeed, Assumption 3.2 is a stricter version of the well-known Slater's condition from convex optimization. It guarantees that $(P_u)$ and $(P_p)$ are well-posed. For overparametrized models, this assumption is not strict. Assumption 3.7, on the other hand, is not straightforward to satisfy at first sight. It is, however, a typical assumption used to derive duality results in convex optimization known as "linear independence constraint qualification" or LICQ [16]. In fact, it is the weakest constraint qualifications for convex optimization. This suggests that it may be possible to replace Assumption 3.7 by a stricter version of Assumption 3.2, although this is not immediate. We therefore choose to state the weakest assumptions under which our results would hold.
>
> We will expand the description of these assumptions already present in Section 3.1 to highlight their mildness and limitations in the revised version.

---

> > ### Comment · Reviewer_Y13X · 2023-11-20
> >
> > I am a bit confused by the references (e.g. ii, iii, (7)) in the response as they don't seem to correspond to the numbering in the review.
> >
> > My main question was how the approach changes with changes in the algorithm it applies to. This is made a bit clearer with the specific discussion of Adam, but I think it could be more clearly laid out. The separation of the errors into algorithm error and approximation error and such are nice.
> >
> > Its a bit of a side topic, but the statement on poorly chosen kernel having approximation error is not quite right. Any Gaussian kernel will have zero approximation error (as will any NN) since such kernels are universal and hence can approximate any function arbitrarily well. It will also have zero algorithm error.  So the issue must lie in the "generalization error". While the current work is interesting, quantifying what "generalization error" means would be even more interesting.
> >
> > Overall, my review is more positive after author response.

---

> > > ### Author Response · Authors · 2023-11-20
> > > **Response to Comment by Reviewer Y13X**
> > >
> > > In our response, the references are divided in: responses to weaknesses (1) and (2), and responses to questions (1)-(8). We will update the latter to (Q1)-(Q8) to make the distinction clearer.
> > >
> > > We agree that quantifying the generalization error is interesting, but this has been done in [7]. In this contribution the authors analyze the generalization properties of empirical constrained learning problems.
> > >
> > > In our paper we focus on what we think is a fundamental missing aspect of the existing literature: Is it possible for dual gradient ascent methods to recover a primal solution that solves a constrained learning problem? Our answer is "not always" but we also show that under (reasonable) assumptions on the structural properties of the learning problem this is possible, indeed. This result is important because dual gradient ascent is the method of choice for solving constrained learning problems.
> > >
> > > Given our convergence results and the generalization error bounds in [7] generalization bounds for dual gradient ascent follow from a triangle inequality.
> > >
> > > Regarding the kernel example and its approximation error, the reviewer is correct. What we meant in the response was "ONE poorly chosen kernel". Gaussian kernels are universal function approximators only asymptotically (for an infinite number of kernels and/or for an RKHS that becomes increasingly less smooth).

---

> > > > ### Comment · Reviewer_Y13X · 2023-11-20
> > > >
> > > > >  What we meant in the response was "ONE poorly chosen kernel". Gaussian kernels are universal function approximators only asymptotically (for an infinite number of kernels and/or for an RKHS that becomes increasingly less smooth)
> > > >
> > > > That isn't quite how kernels work. For example, any given Gaussian kernel defines an infinite set of basis functions centered at data points and which can be used to fit any set of data exactly. Adding more kernels decreases generalization error.

---

> > > > > ### Author Response · Authors · 2023-11-21
> > > > > **Response to Comment by Reviewer Y13X**
> > > > >
> > > > > We understand what you are saying and we are in agreement about how kernels work. We feel, however, that we are going off on a tangent. Our paper, as we have acknowledged, does not look at generalization error.  This has been done in [7]. We are looking at algorithmic error, which, as we hope to have explained well, is a fundamental and challenging problem.

---

> ### Author Response · Authors · 2023-11-20
> **Response to questions 1-6 of reviewer Y13X**
>
> **Questions**
>
> **(Q1)** Due to the correlation between the protected features and other features, simply hiding the protected features from the model does not guarantee fairness. In fact, removing race as a predictor in our example has almost no impact on the predictions of the model (the accuracy and prediction disparity between races remain unchanged). This is a well-studied phenomenon in the fairness literature (see, e.g., [11]). This shows the need for the addition of an explicit constraint in order to guarantee fairness (in our case, counterfactual fairness as defined in [17]).
>
>
> **(Q2)** We thank the reviewer for pointing out this typo. Indeed, when the parametrization used in a neural network, $\mathcal{F}_{\theta}$ is not a convex hypothesis set. That is why, even though the losses $\ell_i$ are convex functionals, $(P_p)$ is not a convex problem. This is in fact one of the major challenges overcome by this challenge (see general response for more details).
>
> As we argue in the general response, this paper does not deal with the approximation (ii) and generalization (iii) errors. It deals with the algorithmic error (iii), i.e., it studies the primal iterates $f_{\theta}(\lambda_t)$ obtained as a by-product of Alg. 1 to understand whether they provide a (feasible) solution of $(P_p)$. This is not straightforward even in the convex setting and prior results in the non-convex case rely on randomization (see, e.g., guarantees in [10], Theorem 2; [11], Theorem 4.1; [7], Theorem 3). We explicitly show that, under mild conditions, randomization is not necessary. In view of the challenges and novel proof techniques deployed in this paper to characterize error (iii) alone (i.e., on whether Alg. 1 can be used to solve $(P_p)$), considerations on the generalizability, robustness, and performance on untrained data of solutions of $(P_p)$ are left for future work.
>
> **(Q3)** This could indeed be done without major modifications since (strong-)convexity and the Lipschitz continuity of the gradients is preserved under expectation (the latter requires some regularity in order to exchange expectation and differentiation, e.g., this can be done using dominated convergence theorem for bounded $\mathcal{F}$). We choose to use Assumption 3.1 and 3.6 since imposing smoothness and convexity on $ \tilde{\ell} $ is stricter. That being said, the reviewer has a point that this assumption is more natural and we will therefore include this comment in the revised version of the manuscript.
>
> **(Q4)** As mentioned in Section 3.1, the unparametrized problem has no duality gap  $ (P_u^\star = D_u^{\star}) $ since it is a convex problem for which Slater's condition holds (Assumptions 3.1-3.2). As the reviewer correctly notes, this is not the case for the parametrized problem. This is in fact one of the major challenges that this work has to overcome (see general responses). Under our Assumption 3.1-3.3, the duality gap of $(P_p)$ can be bounded as in [7], Proposition 3 (see again general response for details). Combining our near-feasibility results from Corollary 3.9 with this duality gap bound yields a guarantee in terms of the objective values $|\ell_0(f_{\theta}(\lambda^*_p)) - P^*|$ (further details are provided in answer 1.3 to reviewer z2M9). We will include these comments and result in the revised version of the manuscript.
>
> **(Q5)** We refer the reviewer to the detailed discussion provided in a previous answer.
>
> **(Q6)** No. As we mention in the beginning of section 3.3, sections 3.3.1 and 3.3.2 provide an outline of the proof of Theorem 3.5 and Corollary 3.9, the main results of this paper. Indeed, Theorem 3.5 is obtained by directly combining Propositions 3.10 and 3.11. We will further clarify this  point in the beginning of section 3.3.

---

> ### Author Response · Authors · 2023-11-20
> **Response to questions 7-8 of reviewer Y13X**
>
> **(Q7)** The reviewer has a point. We will carefully review the notation of the paper and keep $D$ to refer only to the value of the dual problems $(D_u)$ and $(D_p)$. In the quote from the reviewers question, $D$ refers to the dataset (which only appears and is relevant in Section 5). The quantity $D^{\star}_p$ from Lemma 4.1 is defined in Section 2.2 as the optimal value of the parametrized dual problem $(D_p)$.
> Alg. 1, however, is correctly defined in terms of $\ell$. Indeed, recall that this paper does not deal with the estimation errors. Hence, we do not consider empirical approximations of the expectations in $(P_p)$ or $(D_p)$. Only when presenting the numerical experiments in Section 5 do we use these empirical approximations in order to illustrate our results in a concrete application. We refer the reviewer to our general response for a more detailed explanation on the scope and challenges of this paper.
>
> **(Q8)**     Yes, we mean Adam. As per Section 2.1, the constraints in this application are defined as:
> $$
> \mathbb{E}\_{(x, y)} \left[ D_{KL}(f\_{\theta}(x) || f\_{\theta}(\rho_i (x)) \right] \leq c
> $$
> where the $\rho_i$ denote changes in the protected features (in our case gender and race). In the current version of the manuscript the protected features are denoted by $z$, but after addressing comments from reviewer Z2M9, we find that the above presentation is less prone to misunderstanding. In other words, $\ell_i(\phi) = \mathbb{E}\_{(x, y)} \left[ D_{KL}(\phi(x) || \phi(\rho_i (x)) \right]$. As we mention in Section 5, $\ell_0$ is the cross-entropy loss.
>
> Notice that these losses are convex and continuously differentiable. Hence, when using weight decay (which bounds the norm of $\theta$), $\ell_0$ is strongly convex and all losses are Lipschitz continuous and smooth. While constants for these properties ($M$, $\beta$, $\mu_0$) can be estimated, typical estimates are quite loose. Additionally, as we explain in our answer to rUm3, the richness of the parametrization $\nu$ in Assumption 3.3 cannot typically be computed, although for many parametrizations (e.g., NN), it is known that any $\nu > 0$ can be achieved using large enough models. Though we cannot compute the exact value of our bound, we directly observe the behaviors it predicts in our experiments. For instance, Figure 2(right) shows that, as predicted by Corollary 3.9, the constraint violation of the best iterate decreases as the model capacity increases.

---

### Author Response · Authors · 2023-11-20
**General Response**

We thank the reviewers for their time and feedback. In what follows, we clarify an important point regarding the scope of this work.

There are three major sources of errors when solving learning problems: (i) the **estimation error**, that arises from approximating expectations using samples (statistical vs empirical); (ii) the **approximation error**, that arises from using models with limited functional representation capability (all measurable functions vs neural networks); and (iii) the **optimization error**, that arises from the use of numerical algorithms to optimize the model parameters. In (standard) unconstrained learning, errors (i) and (ii) are the leading challenges since convergence properties of unconstrained optimization algorithms are well-understood in convex (see, e.g., [1, 2]) as well as many non-convex setting (e.g., for overparametrized models as in [3,4,5,6]). This is not the case in constrained learning, which is why our paper specifically targets error (iii) (see discussion on errors (i) and (ii) below).

Indeed, two challenges must be overcome in the non-convex case. The first is a lack of strong duality (as pointed out by reviewer rUm3). In other words, the value $D_p^\star$ of the dual problem $(D_p)$ need not be a good approximation of the value $P_p^\star$ of $(P_p)$. This was tackled in [7]. Explicitly, Proposition 3.3 states that under Assumptions 3.1--3.3 in our paper, the duality gap of problem $(P_p)$ is bounded as in:
\begin{equation}
P^*_p - D^*_p \leq M\nu(1+\\| \tilde{\lambda}^* \\|_1 )\text{,}
\end{equation}
where $\tilde{\lambda}^*$ maximizes $\tilde{g}_p(\lambda) = g_p(\lambda)+M \nu \\| \lambda \\|_1$.

This result, however, only show that the dual problem can be used to approximate the *value* of the constrained problem. It says nothing about whether it can provide a feasible solution, which is the issue we address in this paper. Indeed, as pointed out by reviewer z2M9, once the near-feasibility of a Lagrange minimizer $f_{\theta}(\lambda^*_p)$ is established, this bound on the duality gap can be used to prove that it is not only approximately feasible, but also approximately optimal, i.e., to bound $|P^*_p - \ell_0(f\_{\theta}(\lambda^*_p))|$. Note that while $D^*_p$ involves the complete Lagrangian, this new bound is only in terms of $\ell_0$. We elaborate on this point in the responses to reviewers rUm3 and z2M9 and will add it to the revised version.

The other challenge is, therefore, to obtain such a near-feasible Lagrange minimizer. Indeed, even when strong duality holds (e.g., for convex, deterministic problems), dual ascent methods such as Alg. 1 (also referred to as *best response*) need not converge. Even if it did, it may not be possible to extract a *feasible solution* for $(P_p)$ from the optimal dual variables $\lambda^\star_p$. For instance, in linear programs the Lagrangian $L(\phi, \lambda^\star)$ is a constant (independent of $\phi$), i.e., every solution canditate is a Lagrange minimizer ($\Phi^\star(\lambda^\star) = \mathcal{F}$). For convex problems, this can be tackled using primal averaging, i.e., taking the average of the sequence $\theta_t$ [8], or modified GD algorithms, as in the last iterate guarantees from [9]. In the non-convex case (i.e., this paper), existing guarantees require *randomization*, which is far from practical (see, e.g., guarantees in [10], Theorem 2; [11], Theorem 4.1; [7], Theorem 3).

In this paper, we show conditions on the constrained learning problem under which a (near-)feasible dual solution (which is therefore also near-optimal), can be obtained from the *best iterate*, **without the need for randomization**. To the best of our knowledge, similar results exist only in convex settings, e.g., [9].

---

> ### Author Response · Authors · 2023-11-20
> **References.**
>
> [1] Nesterov, Yurii. Introductory lectures on convex optimization: A basic course. Vol. 87. Springer Science & Business Media, 2003.
>
> [2] Bach, F., Jenatton, R., Mairal, J., Obozinski, G., Sra, S., Nowozin, S. and Wright, S.J., 2012. Optimization for Machine Learning.
>
> [3] R. Ge, J. D. Lee, and T. Ma, “Learning one-hidden-layer neural net-works with landscape design,” in International Conference on Learning Representations, 2018.
>
> [4] A. Brutzkus and A. Globerson, “Globally optimal gradient descent for a convnet with gaussian inputs,” in International Conference on Machine Learning, 2017, pp. 605–614.
>
> [5] M. Soltanolkotabi, A. Javanmard, and J. D. Lee, “Theoretical insights into the optimization landscape of over-parameterized shallow neural networks,” IEEE Transactions on Information Theory, vol. 65, no. 2, pp. 742–769, 2018
>
> [6] C. Zhang, S. Bengio, M. Hardt, B. Recht, and O. Vinyals, “Understanding deep learning requires rethinking generalization,” in International Conference on Learning Representations, 2017
>
> [7] Chamon, Luiz FO, et al. "Constrained learning with non-convex losses." IEEE Transactions on Information Theory 69.3 (2022): 1739-1760.
>
> [8] Nedić, A. and Ozdaglar, A., 2009. Approximate primal solutions and rate analysis for dual subgradient methods. SIAM Journal on Optimization, 19(4), pp.1757-1780.
>
> [9] Daskalakis, C. and Panageas, I., 2018. Last-iterate convergence: Zero-sum games and constrained min-max optimization. arXiv preprint arXiv:1807.04252.
>
> [10] Cotter, A., Jiang, H., Gupta, M.R., Wang, S.L., Narayan, T., You, S. and Sridharan, K., 2019. Optimization with Non-Differentiable Constraints with Applications to Fairness, Recall, Churn, and Other Goals. J. Mach. Learn. Res., 20(172), pp.1-59.
>
> [11] Kearns, M., Neel, S., Roth, A. and Wu, Z.S., 2018, July. Preventing fairness gerrymandering: Auditing and learning for subgroup fairness. In International conference on machine learning (pp. 2564-2572). PMLR.
>
> [12] Shor, N.Z., 1998. Nondifferentiable optimization and polynomial problems (Vol. 24). Springer Science & Business Media.
>
> [13] Hornik, K., Stinchcombe, M. and White, H., 1989. Multilayer feedforward networks are universal approximators. Neural networks, 2(5), pp.359-366.
>
> [14] Mitzenmacher, M. and Upfal, E., 2017. Probability and computing: Randomization and probabilistic techniques in algorithms and data analysis. Cambridge university press
>
> [15] Chamon, L. and Ribeiro, A., 2020. Probably approximately correct constrained learning. Advances in Neural Information Processing Systems, 33, pp.16722-16735.
>
> [16] Bertsekas, D.P., 1997. Nonlinear programming. Journal of the Operational Research Society, 48(3), pp.334-334.
>
> [17] Kusner, M.J., Loftus, J., Russell, C. and Silva, R., 2017. Counterfactual fairness. Advances in neural information processing systems, 30.
>
> [18] Shalev-Shwartz, S. and Ben-David, S., 2014. Understanding Machine Learning: From Theory to Algorithms.
>
> [19] Boyd, S., Xiao, L. and Mutapcic, A., 2003. Subgradient methods. lecture notes of EE392o, Stanford University, Autumn Quarter, 2004, pp.2004-2005
>
> [20] Grimmer, B., 2019. Convergence rates for deterministic and stochastic subgradient methods without Lipschitz continuity. SIAM Journal on Optimization, 29(2), pp.1350-1365.
>
> [21] Clerk’s Office, Broward County Sherrif’s Office, Florida Department of Corrections, COMPAS Recidivism Risk Score Data and Analysis.{ ProPublica 2020.
>
> [22] Diana, E., Gill, W., Kearns, M., Kenthapadi, K. and Roth, A., 2021, July. Minimax group fairness: Algorithms and experiments. In Proceedings of the 2021 AAAI/ACM Conference on AI, Ethics, and Society (pp. 66-76).
>
> [23] Bazaraa, M.S., Sherali, H.D. and Shetty, C.M., 2013. Nonlinear programming: theory and algorithms. John wiley & sons.
>
> [24] Larsson, T., Patriksson, M. and Strömberg, A.B., 1999. Ergodic, primal convergence in dual subgradient schemes for convex programming. Mathematical programming, 86, pp.283-312.

---

### Meta-Review · Area_Chair_m453 · 2023-12-06

**Metareview:**

This paper explores the issue of constrained learning within a dual-ascent framework. It focuses on bounding the infeasibility of the solution found by the dual method. The methodology involves an infinite-dimensional, non-parameterized version of the dual problem to establish these bounds.

The main strength is that the approach allows for estimating the error in infinite-dimensional primal problems, and the findings are then applied to determine the error in the actual finite-dimensional primal solution using dual ascent.

The main weakness is the lack of study on the objective value in addition to the infeasibility.

**Justification For Why Not Higher Score:**

The main weakness is the lack of study on the objective value in addition to the infeasibility.

**Justification For Why Not Lower Score:**

The main strength is that the approach allows for estimating the error in infinite-dimensional primal problems, and the findings are then applied to determine the error in the actual finite-dimensional primal solution using dual ascent.

---

### Decision · Program_Chairs · 2024-01-16

Accept (poster)